# SVIP is a molecular determinant of lysosomal dynamic stability, neurodegeneration and lifespan

Alyssa E. Johnson[1,5], Brian O. Orr[1], Richard D. Fetter [1], Armen J. Moughamian[1,2], Logan A. Primeaux[3], Ethan G. Geier[3,4], Jennifer S. Yokoyama[4], Bruce L. Miller[4] & Graeme W. Davis [1✉]

Missense mutations in Valosin-Containing Protein (*VCP*) are linked to diverse degenerative diseases including IBMPFD, amyotrophic lateral sclerosis (ALS), muscular dystrophy and Parkinson's disease. Here, we characterize a VCP-binding co-factor (SVIP) that specifically recruits VCP to lysosomes. SVIP is essential for lysosomal dynamic stability and autophagosomal–lysosomal fusion. *SVIP* mutations cause muscle wasting and neuromuscular degeneration while muscle-specific *SVIP* over-expression increases lysosomal abundance and is sufficient to extend lifespan in a context, stress-dependent manner. We also establish multiple links between *SVIP* and *VCP*-dependent disease in our *Drosophila* model system. A biochemical screen identifies a disease-causing VCP mutation that prevents SVIP binding. Conversely, over-expression of an *SVIP* mutation that prevents VCP binding is deleterious. Finally, we identify a human *SVIP* mutation and confirm the pathogenicity of this mutation in our *Drosophila* model. We propose a model for VCP disease based on the differential, co-factor-dependent recruitment of VCP to intracellular organelles.

[1] Department of Biochemistry and Biophysics, Kavli Institute for Fundamental Neuroscience, University of California, San Francisco, San Francisco, CA 94158, USA. [2] Department of Neurology, University of California, San Francisco, San Francisco, CA 94158, USA. [3] Department of Biological Sciences, Louisiana State University, Baton Rouge, LA 70803, USA. [4] Department of Neurology, Memory and Aging Center, University of California, San Francisco, San Francisco, CA 94158, USA. [5] Present address: Department of Biological Sciences, Louisiana State University, Baton Rouge, LA 70803, USA. ✉email: Graeme.davis@ucsf.edu

The *Valosin-Containing Protein* (*VCP*) gene encodes an AAA+ ATPase[1] that has been linked to a broad spectrum of autosomal dominant degenerative diseases including IBMPFD (inclusion body myopathy with Paget's disease and fronto-temporal dementia; Watts et al.[2]), amyotrophic lateral sclerosis (ALS)[3,4], spastic paraplegia[5], muscular dystrophy[6], and Parkinson's disease[7,8]. Patients with IBMPFD generally present to the clinic with muscle weakness in mid to late life[4,9,10]. Approximately one-third of patients have degeneration of the central nervous system including the fronto-temporal lobes[4,9].

The *VCP* gene is cell lethal when deleted[11,12]. As a likely consequence, disease-causing *VCP* mutations are primarily distributed in the N-terminal regulatory region, a substrate/co-factor binding domain. To date, despite a wealth of information regarding VCP activity, there is no clear consensus regarding how mutations in the VCP co-factor binding domain cause disease pathogenesis. For example, there is considerable debate regarding whether disease-causing mutations are gain-of-function versus loss-of-function mutations[7,13–16]. Furthermore, although defects in protein clearance mechanisms appear to be a root cause of *VCP*-related disease[17], consistent with the general characterization of VCP as a ubiquitin-dependent protein segregase[18,19], there remains debate regarding which protein clearance pathways are compromised given that VCP participates in ER-associated degradation (ERAD)[20], autophagy[15,16], mitophagy[7,21], and ubiquitin-proteasome-associated degradation (UPS)[18]. Although pathogenic mutations in VCP do not seem to impair the UPS or ERAD protein degradation pathways[13,14], impaired autophagy[14], mitochondrial quality control[22,23], and lysosomal dysfunction[16] have been proposed as cell biological consequences of disease-causing *VCP* mutations.

One possibility is that VCP is an essential enzymatic resource that is dynamically deployed to different organelle systems, as necessary, to resolve sites of local cellular stress (Fig. 1a). In this model, the critical determinants of cellular robustness are the co-factors that bind VCP and dictate the distribution of VCP throughout the cell. According to this model, disease-causing mutations in the co-factor binding domain could, conceivably, result in both gain and loss-of-function effects if a fixed amount of VCP enzyme becomes errantly redistributed within the cell in the absence of binding to a specific co-factor. These effects might be exacerbated under conditions of cellular stress and accelerate during disease progression. With this model in mind, we set out to identify and characterize the function of VCP-binding proteins, focusing on currently unknown mechanisms that localize VCP to the lysosome.

We previously identified an essential activity for VCP at *Drosophila* muscle lysosomes[16]. We demonstrated that muscle lysosomes form a dynamic, tubular lattice that extends throughout individual muscle cells (insect and mammalian muscle). When VCP is mutated, or when VCP is acutely depleted from the cytosol, tubular lysosomes fragment, and autophagosome–lysosome fusion is blocked, ultimately leading to impaired autophagy, impaired protein clearance, and mitochondrial defects[16]. Fragmentation of muscle lysosomes is one of the earliest cellular defects known to occur following the disruption of VCP[16]. Notably, patients with disease-causing mutations in *VCP* present to the clinic with primary muscle weakness[10], highlighting the potential relevance of VCP-dependent control of muscle lysosomes and autophagy. We sought to identify co-factor(s) that bind and recruit VCP to the newly identified, highly dynamic system of muscle tubular lysosomes.

In this study, we define the function of a VCP-interacting co-factor termed SVIP (Small VCP-Interacting Protein)[24,25]. SVIP was initially identified as a VCP-interacting protein and a negative regulator of ERAD[24,25]. Prior work in immortalized human cells (HeLa) demonstrated that SVIP over-expression is sufficient to alter the subcellular distribution of VCP to "peri-nuclear granules" and the plasma membrane[26]. SVIP knockdown correlated with altered LC3 lipidation in these cells. However, beyond this, little is known about SVIP function.

Here we define SVIP as an essential VCP co-factor that recruits VCP to lysosomes, an activity that is essential for the dynamic stability of the muscle tubular-lysosomal lattice. Loss of SVIP disrupted the muscle lysosomal network, impaired autophagosome–lysosome fusion, and caused myriad degenerative defects including muscle wasting, neuromuscular degeneration, dendrite degeneration in motoneurons, progressive motor dysfunction and decreased organismal lifespan. We link SVIP to VCP-dependent degenerative disease through phenotypic analysis of a known human disease-causing mutation in *Drosophila* VCP, and we present the identification of an *SVIP* mutation in a human patient diagnosed with fronto-temporal dementia (FTD). We subsequently confirmed the pathogenicity of this human *SVIP* mutation in our *Drosophila* model. Although we cannot conclude causality, this human mutation and our subsequent characterization in vivo in *Drosophila* argues that the function of *SVIP* has relevance to the human condition. Taken together, our data define a model for VCP disease based on the differential, co-factor-dependent recruitment of VCP to intracellular organelles. Further, our data underscore the relevance of lysosome dysfunction as an underlying cause for VCP-linked diseases of the peripheral and central nervous systems.

## Results

**Drosophila SVIP recruits VCP to tubular lysosomes**. We identified a small ORF (CG32039) that shares high sequence homology to the human, mouse, and rat SVIP proteins (Fig. 1b). Critical residues are conserved in CG32039 including glycine 2 that is required for SVIP myristoylation and membrane tethering, as well as the region encompassing a putative VCP-interaction motif (VIM) (Fig. 1b). Thus, based on sequence homology, CG32039 (hereafter referred to as SVIP) is a strong candidate to be the *Drosophila* SVIP ortholog.

We first demonstrate that *Drosophila* SVIP directly binds to VCP. Purified recombinant GST-SVIP and MBP-VCP$^{N-D1}$ (containing the N-terminal and first D1 ATPase domain) proteins directly bind and this interaction was lost when two arginine residues (R24, 25) in the VIM sequence of SVIP (SVIP$^{RR}$) were mutated to alanine (Fig. 1c). Additionally, mutation of VCP aspartate 32, a residue previously demonstrated to disrupt VCP interaction with VIM motifs[27] also disrupted the SVIP–VCP interaction in vitro (Supplementary Fig. S1A, B). Thus, we define SVIP as a VCP-binding protein.

In order to define the subcellular distribution of SVIP, we took two approaches. First, we generated an antibody against GST-SVIP to examine the endogenous expression patterns of *Drosophila* SVIP. SVIP is expressed in both 3rd instar larvae and adults, being expressed at relatively lower levels in the brain compared to the rest of the body (Supplementary Fig. S1C). Second, we used CRISPR to knock-in a superfold GFP (sfGFP) tag at the C terminus of the endogenous *SVIP* gene (see "Methods" section). This approach allows us to assess SVIP localization in live tissue. We demonstrate that SVIP is expressed in the body wall muscles of 3rd instar larvae where it co-localizes with a lysosomal marker Spin-RFP (Fig. 1d, e).

We previously demonstrated that muscle lysosomes form a dynamic tubular lattice that extends throughout the muscle cytoplasm[16]. This dynamic lattice can be observed by imaging an RFP tagged integral lysosomal membrane protein (Spin-RFP;

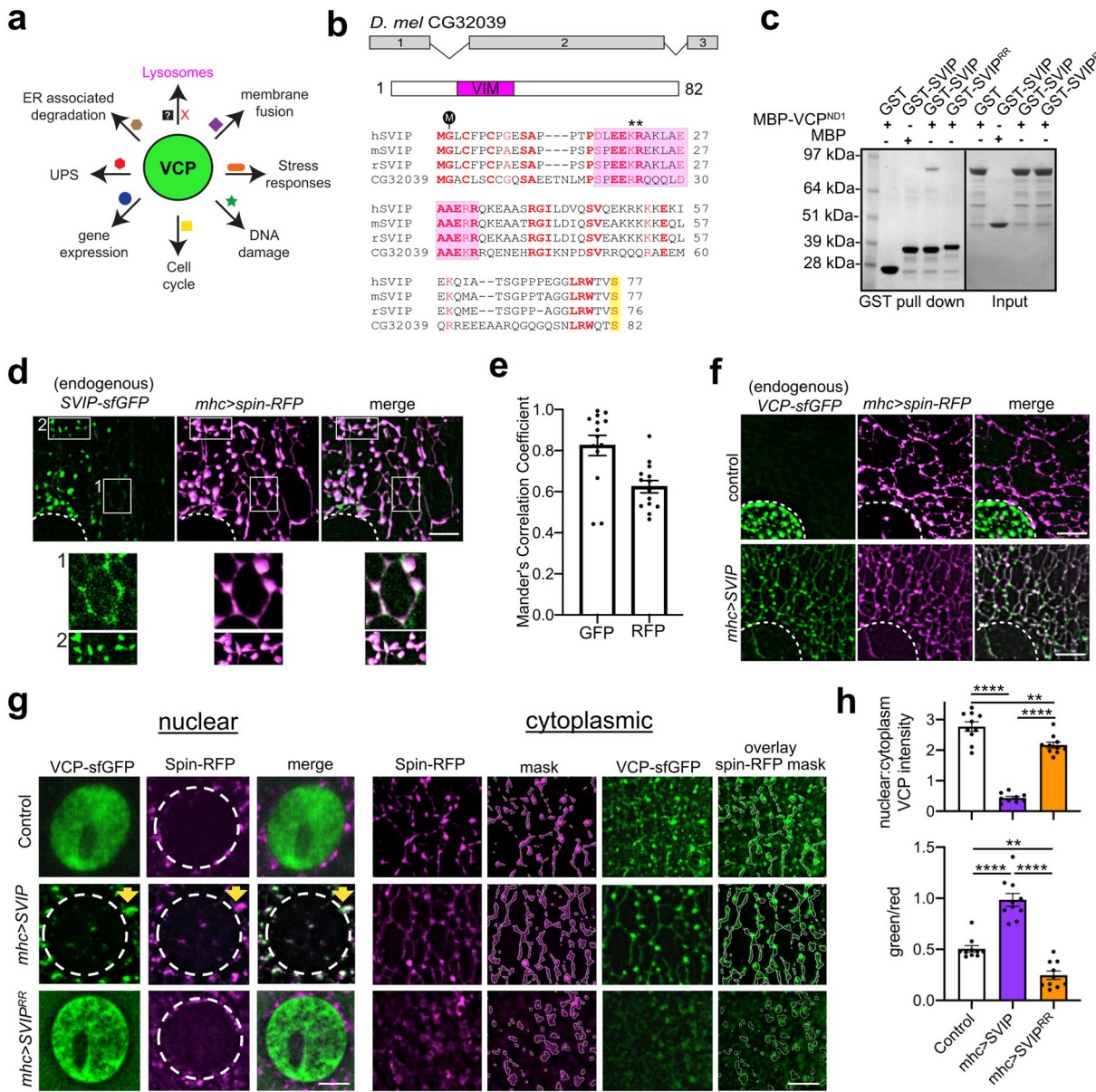

**Fig. 1 *Drosophila* SVIP directly recruits VCP to tubular lysosomes in muscles. a** Distinct co-factors (shapes of different colors adjacent to arrows) spatially regulate VCP subcellular distribution and function. **b** Schematic of *Drosophila* CG32039 gene and protein organization (top) and sequence alignment with human, mouse, and rat SVIP orthologs (bottom). The region shaded in magenta corresponds to the predicted VCP-interaction motif (VIM), M indicates myristoylation site, asterisks indicate arginine residue mutated in part C, and serine highlighted in yellow indicates SVIP patient mutation identified (Fig. 9). **c** In vitro binding of recombinant MBP-VCP and GST-SVIP fusion proteins. **d** Co-localization of endogenously tagged SVIP-sfGFP and Spin-RFP (lysosomes) in *Drosophila* muscles. The dotted line indicates the nucleus. Boxes indicated insets, below at higher magnification. **e** Mander's correlation coefficients for SVIP-sfGFP (GFP) and Spin-RFP (RFP) signal ($n = 13$ muscle cells examined over five independent animals). **f** Co-imaging of VCP-sfGFP and Spin-RFP in control and $SVIP^+$ over-expression in muscles. Dotted lines as in **d**. **g** Co-imaging of endogenously tagged VCP-sfGFP and UAS-Spin-RFP (lysosomes) in control compared to $SVIP^+$ and $SVIP^{RR}$ over-expression in muscles. Dotted lines as in **d** and **f**. Yellow arrow (middle row) highlights signal co-localization. **h** Nuclear to cytoplasmic ratios of VCP-sfGFP signal was quantified for the genotypes indicated (top graph). Spin-RFP signal was used to create a masked ROI (white lines, "mask") and plots represent GFP/RFP signal intensities within the masked region (bottom graph, see "overlay" in **g**). Data, artificial imaging units. ($n = 10$ muscle cells examined over five independent animals). See also Supplementary Fig. S1. Scale bars 5 μm. Data presented as mean and SEM. Student's *t*-test for individual comparisons (*$p < 0.05$; **$p < 0.01$; ***$p < 0.001$, ****$p < 0.0001$).

Supplementary Movie 1). This tubular-lysosomal system is present in a diversity of cells and organisms, inclusive of cultured mouse myoblasts (ref. [16] and unpublished observation). Importantly tubular lysosomes are only visible in live tissue, collapsing into vesicular structures by a diverse array of fixation methods[16]. Here we demonstrate that SVIP localizes at lysosome tubules that are distributed throughout the cytosol (see Fig. 1d, insets 1 and 2 as well as quantification in Fig. 1e).

To determine whether SVIP recruits VCP to muscle tubular lysosomes, we tested whether over-expressing *Drosophila* SVIP drives enhanced VCP localization to lysosomes. Because muscle lysosomes do not retain their dynamic tubular structure following fixation, we generated two transgenic animals to allow us to over-express SVIP and monitor endogenously tagged VCP protein in vivo. First, we created a *UAS-SVIP* transgenic animal enabling SVIP over-expression. Second, we used CRISPR technology to tag

VCP with sfGFP at its endogenous locus. Using these tools, we demonstrate that endogenous VCP protein concentrates on the nuclear membranes in muscle and is weakly distributed throughout the rest of the muscle cell. Image analysis of extra-nuclear VCP reveals a significant concentration at tubular lysosomes (Fig. 1f, g; note that exposure time for VCP-sfGFP is extended in (g) compared to (f) to enable visualization of lysosome-associated VCP). Strikingly, when *UAS-SVIP* is over-expressed using the muscle-specific *mhc-GAL4* driver, endogenously expressed VCP-sfGFP re-localizes to tubular lysosomes throughout the muscle cell (Fig. 1f, g). The SVIP-dependent re-localization of VCP causes depletion of VCP from the peri-nuclear compartment where VCP is enriched at a steady state (Fig. 1f, g). As a control, we over-expressed *UAS-SVIP^{RR}*, an SVIP variant that cannot bind VCP (Fig. 1c). As expected, *SVIP^{RR}* over-expression did not induce VCP re-localization to the tubular-lysosomal network (Fig. 1f, g). We quantified these effects by measuring two parameters. First, we quantified the juxta-nuclear to cytoplasmic (representative of lysosomal tubules located throughout cytoplasm) ratio of endogenously tagged VCP-sfGFP (Fig. 1h). Second, we used the Spin-RFP channel to create a mask of the tubular-lysosomal network and quantified VCP-sfGFP intensity in this region of interest (Fig. 1g, h). Collectively, these data demonstrate that *Drosophila* SVIP binds VCP and is sufficient to re-localize endogenous VCP to the tubular-lysosomal system.

**SVIP-dependent VCP recruitment is essential to stabilize lysosomal structure and function**. To investigate the function of *SVIP* in vivo, we generated an *SVIP* mutant using the CRISPR–Cas9 system. We designed a gRNA to direct Cas9 to cut within the first exon of *SVIP*, which resulted in a frame-shift mutation and a pre-mature stop codon upstream of the VIM site (Fig. 2a). Using our SVIP antibody, we validated that the *SVIP* mutation is a protein null by western analysis (Fig. 2a; hereafter referred to as *SVIP^{KO}*).

We previously demonstrated that *VCP* mutations cause fragmentation of the tubular lysosome system in *Drosophila* muscle[16]. If SVIP is necessary to recruit VCP to lysosomes, then the loss of *SVIP* should phenocopy the *VCP* mutant. Prior live imaging studies demonstrate that lysosome tubules are highly dynamic with constant tubule extension, scission, and fusion throughout the muscle cytoplasm[16]. First, we repeat this live imaging (Supplementary Movie 1). Next, we demonstrate that the *SVIP^{KO}* causes fragmentation of the network of lysosome tubules in body wall muscles of 3rd instar larvae (Fig. 2b). Two measures were used to quantify effects on the lysosomal tubular network: (1) the number of junctions per tubule, which assesses the complexity of the tubule network, and (2) the number of tubules within a 100 μm$^2$ region of interest, as a measure of tubule density. Both measures are significantly decreased in the *SVIP^{KO}* (Fig. 2b, pink). This effect requires the SVIP–VCP interaction because over-expression of wild-type *UAS-SVIP* in the *SVIP^{KO}* fully rescues the tubular lysosome system (Fig. 2b, rescue, green), whereas over-expression of the *UAS-SVIP^{RR}* mutant that abolishes SVIP–VCP binding does not rescue (Fig. 2b, *SVIP^{RR}*, orange). Finally, we note that over-expression of *UAS-SVIP* not only restores the lysosomal network but increased tubule density beyond wild-type levels ($p < 0.001$), suggesting that SVIP levels may be limiting for the dynamic stability of the lysosomal tubule network.

We also extend these observations to the adult *Drosophila* abdominal muscles, demonstrating that the *SVIP^{KO}* disrupts tubular-lysosomal networks (Fig. 2c). We demonstrate a significant drop in tubule complexity (junctions per tubule), but

there was only a trend toward fewer tubules. On closer inspection of the collapsed lysosomal network, we find that tubules can still emanate from vesicular lysosomes (Fig. 2c, white arrows). Thus, in the absence of SVIP-dependent recruitment of VCP, it appears that lysosomes retain the ability to generate de novo tubules at a basal rate, but tubules appear unable to fuse with other lysosomes and cannot sustain a stable lattice.

We previously demonstrated that VCP is required for the fusion of autophagosomes with lysosomes, a process that sustains the integrity of the tubular-lysosomal system[16]. To test the involvement of SVIP in VCP-mediated autophagosome–lysosome fusion, we used a dual fluorescent autophagy sensor: mCherry-GFP-Atg8a. Atg8a resides in the membrane of mature autophagosomes. When autophagosomes fuse with lysosomes, GFP fluorescence is quenched by low lysosomal pH, but the mCherry signal is unaffected. Therefore, the efficiency of autophagosome–lysosome fusion can be measured by the amount of GFP fluorescence compared to the amount of mCherry signal. Here, we demonstrate that *SVIP^{KO}* animals display significantly increased GFP fluorescence compared to wild type (Fig. 2d; $p < 0.001$). Thus, the ability of autophagosomes to fuse with lysosomes is defective. Consistent with such a defect, *SVIP^{KO}* animals also accumulated detergent-insoluble protein aggregates (Supplementary Fig. S2). Collectively, these data argue that SVIP-mediated VCP recruitment to muscle lysosomes promotes both homotypic lysosome fusion (to sustain the tubular-lysosomal lattice) and heterotypic membrane fusion with autophagosomes (Fig. 2d).

**Synthetic reconstitution of VCP-lysosomal targeting rescues lysosomes in *SVIP^{KO}***. To further address our emerging model, we asked whether synthetically driving VCP to lysosomes in the *SVIP^{KO}* is sufficient to rescue the tubular lysosome lattice. To do so we designed a system based on FKBP-FRB (FK506 Binding Protein—FRB) dimerization to artificially drive VCP to lysosomes. A variant of FKBP (FKBP$^{F37V}$) that binds to the rapamycin analog B/B dimerizer was fused to the N terminus of the lysosome-specific membrane protein Spinster[16,28]. The complementing FRB tag was fused to the N terminus of VCP (Fig. 2e). To visualize the two proteins, mCherry and Venus tags were fused to the C terminus of Spinster and VCP, respectively (Fig. 2e). In *SVIP^{KO}* animals, in the absence of the B/B dimerizer, lysosomes were vesicular and VCP appeared diffuse throughout the cytoplasm (Fig. 2f, top panels, control). However, after 4 h of B/B incubation, VCP re-localized to lysosomes, and the tubular-lysosomal lattice began regenerating (Fig. 2f, bottom panels, dimerizer). We find that the increase in tubule density in the presence of dimerizer is highly significant ($p < 0.001$), and there is a strong trend toward increased tubule complexity ($p = 0.09$). Thus, synthetic recruitment of VCP to lysosomes, performed in the absence of *SVIP* which normally localizes VCP to lysosomes, is sufficient to enhance the stability of a dynamic lysosomal network. The inability to fully generate a network, inclusive of significant increases in the number of junctions per tubule could be due to the short time (4 h) of dimerizer or reflect a slight impairment of reconstituted VCP activity due to the engineered dimerizer system.

**SVIP functions in muscle, affecting organismal motility, and lifespan**. We sought to assess the organismal effects of *SVIP^{KO}* and define which tissues require SVIP function. In the 3rd instar larvae, we observed no apparent impairment of motility (Supplementary Fig. S3A) and no hallmarks of motor neuron or muscle degeneration (Supplementary Fig. S3B). However, we found that 28% of *SVIP^{KO}* pupa failed to eclose to the adult stage,

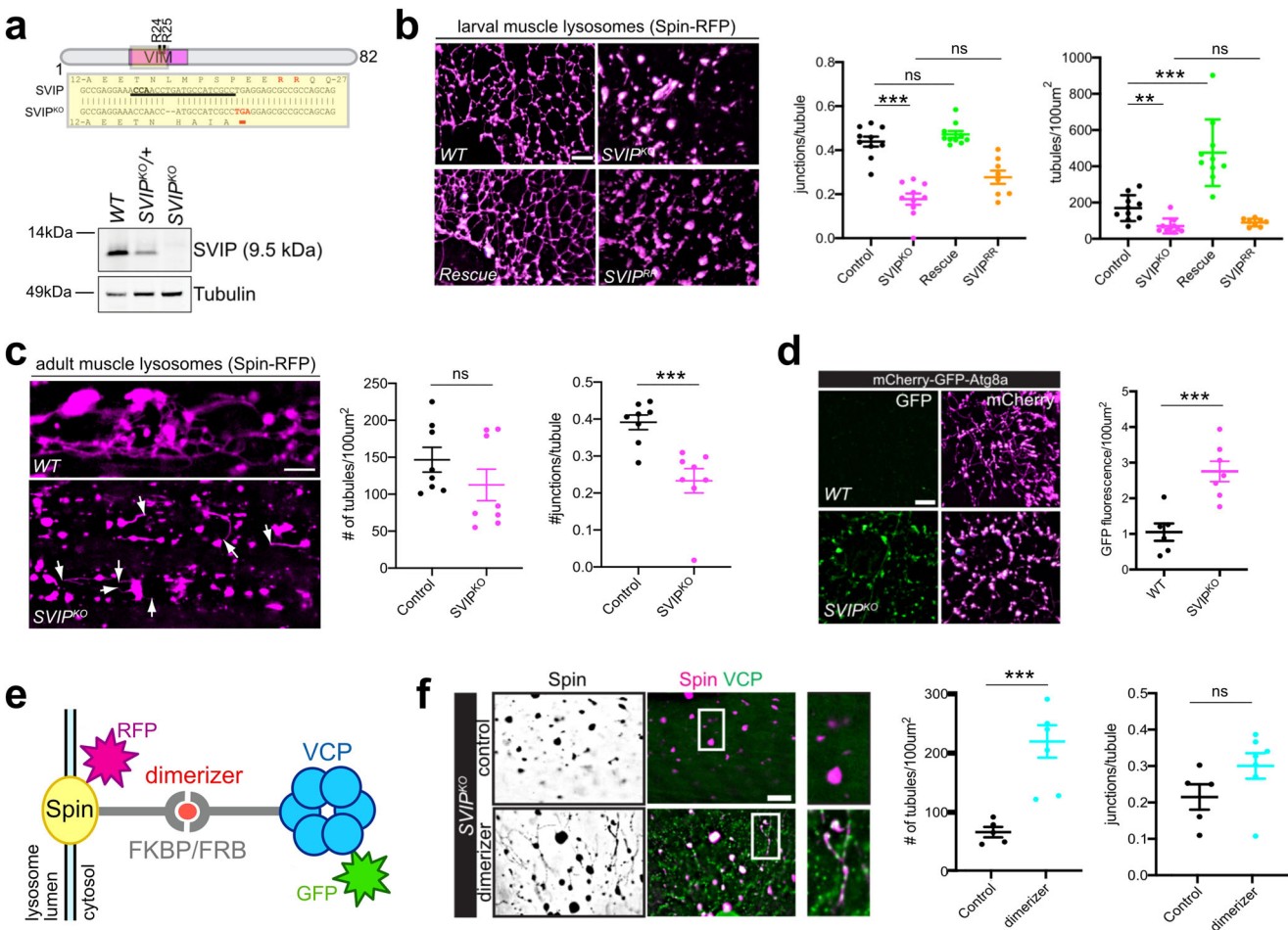

**Fig. 2 SVIP-dependent VCP recruitment to lysosomes is required for lysosome integrity and function. a** Schematic of *SVIP^KO* mutant generated by CRISPR (top) and western blot analysis of SVIP protein levels in *wild type* (*WT*), *SVIP^KO* heterozygous and *SVIP^KO* homozygous mutants (bottom). **b** Representative images of Spin-RFP in 3rd instar larval muscles for the indicated genotypes. Quantification (right) for indicated genotype for tubule complexity (junctions per tubule) and tubule density (tubules/100 μm²). (*n* = 10 muscle cells examined over five independent animals). **c** Representative images of Spin-RFP in 1-week-old adult abdominal muscles for the indicated genotypes. Quantification as in **b**. (*n* = 8 muscle cells examined over four independent animals). **d** Representative images of mCherry-GFP-Atg8a (autophagy sensor) in *wild type* (*WT*) and *SVIP^KO* larval muscles. Quantification at right. (*n* = 7 muscle cells examined over three independent animals). **e** Cartoon diagram of FKBP-FRP dimerization system (components as indicated). **f** Representative images of FKBP-Spin-mCherry and FRB-VCP-Venus localization in the absence (control) or presence of B/B dimerizer. Quantification at right as in **b**. (*n* = 6 muscle cells examined over three independent animals). Scale bars, 5 μm. See also Supplementary Fig. S2. Data presented as mean and SEM. Student's *t*-test for individual comparisons (**p* < 0.05; ***p* < 0.01; ****p* < 0.001). ANOVA for multiple comparisons (**p* < 0.05; ***p* < 0.01; ****p* < 0.001).

compared to just 7% of wild-type animals (Supplementary Fig. S3C). These data argue that SVIP function is relevant to organismal development and may be relevant to health and lifespan. Thus, we turned to the analysis of adult animals where these parameters can be addressed.

In adult *Drosophila*, we find that *SVIP^KO* animals have a reduced lifespan compared to controls (29% decrease in median lifespan) (Fig. 3a). We next asked where SVIP is required for a normal lifespan. Remarkably, diminished lifespan can be fully rescued by muscle-specific expression of *UAS-SVIP* (Fig. 3a). By contrast, neuronal expression of *UAS-SVIP* in the *SVIP^KO* did not rescue viability (Supplementary Fig. S4A). Since SVIP is necessary and sufficient in skeletal muscle to control autophagosomal fusion and the dynamic stability of the lysosomal tubular network, we propose that these activities of SVIP, in muscle, are necessary for normal organismal lifespan.

Next, we asked whether the decrease in lifespan is preceded by impaired motor behavior. Motility was assessed using a commonly employed negative geotaxis assay, taking advantage of a powerful innate behavior in which animals climb the side of a

vial immediately upon being knocked to the bottom. We observe a profound deficit in motor ability in *SVIP^KO* animals that is first apparent at 14 days and progresses to a complete inability to climb the sides of a vial by 28 days (Fig. 3b). The *SVIP^KO* at 28 days are fully viable and do not begin to show increased mortality for another 10 days (Fig. 3a). Thus, progressive motor dysfunction precedes early lethality, consistent with age-related, progressive degeneration in the motor system.

Next, we asked where SVIP is required in order to sustain normal motor behavior. To do so, we assessed climbing performance at 28 days. This time point was chosen as a moment when the *SVIP^KO* is almost completely immobile, but when there remains no change in animal viability. Muscle-specific expression of *UAS-SVIP* in the *SVIP^KO* restores mobility to nearly wild-type values (~90% rescue of negative geotaxis, Fig. 3c), whereas pan-neuronal expression of *UAS-SVIP* in the *SVIP^KO* provides only partial (~50%) rescue (Supplementary Fig. S4B). Thus, while there is some benefit to the expression of SVIP in the nervous system, it is apparent that wild-type animal viability and motility can be achieved when SVIP is expressed exclusively in muscle.

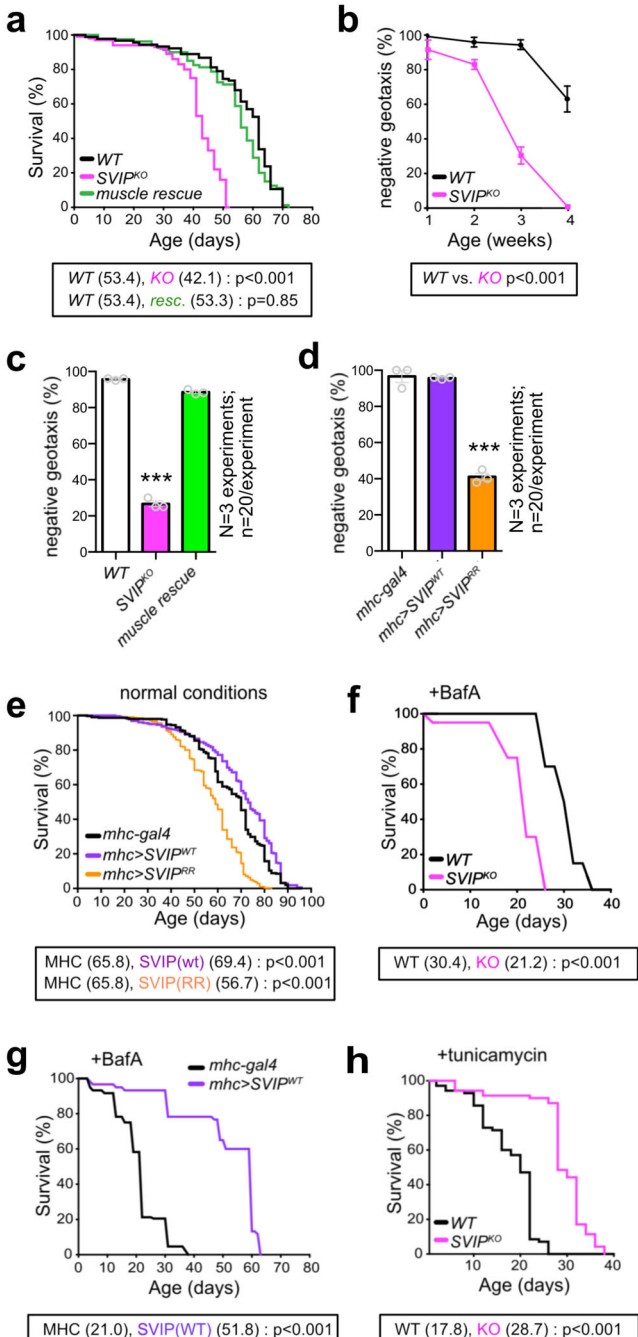

**Fig. 3 SVIP is a stress-dependent survival factor in muscle. a** Lifespan of *WT* (*n* = 90), *SVIP^KO^* (*n* = 100), and *SVIP^KO^* muscle rescue (*n* = 79). **b** Negative geotaxis assay for *WT* and *SVIP^KO^* mutants at the ages indicated. Data from three independent replicates with 20 animals each are presented as mean and SEM for each age point ($p < 0.01$ for 1 and 2 weeks, $p < 0.001$ for 3 and 4 weeks). **c**, **d** Negative geotaxis assay at 4 weeks of age for the genotypes indicated. Data from three independent replicates with 20 animals each are presented as mean and SEM. **e** Lifespan of for control (*mhc-gal4*, *n* = 300) and muscle over-expression of *UAS-SVIP^WT^* (*n* = 298) and *UAS-SVIP^RR^* (*n* = 302). **f** Lifespan of *WT* and *SVIP^KO^* mutants that were constantly fed the lysosome inhibitor, bafilomycin-A (*n* = 60 for both genotypes). **g** Lifespan quantified for control (*mhc-gal4*) and animals with muscle-specific over-expression of *SVIP^WT^* (*mhc-gal4, UAS-SVIP*) that were constantly fed the lysosome inhibitor, BafA (*n* = 69 for each genotype). **h** Lifespan of *WT* and *SVIP^KO^* mutants that were constantly fed the ER inhibitor, tunicamycin (*n* = 70 for each genotype). See also Supplementary Figs. S3 and S4. Mean lifespans are in parentheses. Log-rank tests were used to determine statistical significance. Student's *t*-test for individual comparisons (\**p* < 0.05; \*\**p* < 0.01; \*\*\**p* < 0.001). ANOVA for multiple comparisons (\**p* < 0.05; \*\**p* < 0.01; \*\*\**p* < 0.001).

(Fig. 3e, purple). Finally, over-expression of the SVIP mutant that is unable to bind VCP (SVIP^RR^), significantly diminished lifespan, as predicted (Fig. 3e, orange).

What happens when animals are stressed by the administration of drugs that disrupt the lysosome or impair normal proteostasis? Would *SVIP* muscle over-expression have a beneficial effect? We raised wild type and *SVIP^KO^* animals on food containing the lysosomal inhibitor bafilomycin-A (BafA). Under this condition, wild-type mean lifespan is decreased from ~53 days (Fig. 3a) to ~30 days (Fig. 3f), highlighting the deleterious effects of this drug delivered at concentrations previously used for feeding experiments in *Drosophila*[29]. Importantly, we find that lifespan is even more severely affected in the *SVIP^KO^* compared to wild type (Fig. 3f, pink), with mean lifespan diminished to ~20 days. Thus, loss of SVIP sensitizes the organism to the effects of BafA, as might be expected given that BafA targets the lysosome.

Next, we tested whether muscle-specific SVIP over-expression might counter the deleterious effects of BafA. Indeed, muscle over-expression of *SVIP* in wild-type animals nearly doubles lifespan in the presence of BafA (Fig. 3g; *mhc > SVIP^WT^*). Thus, SVIP over-expression, which is sufficient to increase lysosomal density in muscle, has a beneficial effect on lifespan under conditions of lysosomal stress.

Finally, we asked whether SVIP over-expression is beneficial in the context of other pharmacological stressors. We reared animals on food containing Tunicamycin to stress the ER-resident unfolded protein response. A concentration of Tunicamycin was chosen based on prior work[30,31] and which induces elevated GRP78/Bip levels in both WT and *SVIP^KO^* animals (Supplementary Fig. S4C). As predicted, wild-type mean lifespan is diminished, dropping from ~53 days (Fig. 3a) to ~17 days (Fig. 3h). As above, we repeated this assay in the *SVIP^KO^* animals. Remarkably, knockouts live dramatically longer than wild type in the presence of Tunicamycin (Fig. 3h, pink). Several possible explanations exist. Among these, *SVIP^KO^* animals could be Tunicamycin resistant because increased amounts of VCP can be deployed to the ER in the absence of SVIP-mediated localization of VCP to the lysosome. This is consistent with the initial identification of SVIP as a negative regulator of ER-associated protein degradation[24]. These data support our initial model in which the dynamic, co-factor-dependent deployment of VCP to different organellar systems can have powerful effects on organismal health and lifespan (Fig. 1a).

To further confirm the essential activity of muscle SVIP, and to link SVIP activity to the muscle-specific function of VCP, we compared the effects of muscle over-expression of wild-type *UAS-SVIP* versus mutant *UAS-SVIP^RR^* that is unable to bind VCP. Muscle-specific over-expression of *UAS-SVIP^WT^* was without effect, but muscle-specific over-expression of *UAS-SVIP^RR^* had a profound (~65%) decrease in animal mobility in our negative geotaxis assay (Fig. 3d). These data are consistent with SVIP-dependent recruitment of VCP to muscle lysosomes being essential.

Next, we returned to the effects of SVIP on organismal lifespan. Given the potent activity of SVIP in muscle, we asked whether muscle over-expression of *SVIP*, in a wild-type background, would further extend lifespan. Indeed, lifespan extension is statistically significant when *SVIP* is expressed in muscle, although the absolute magnitude of the effect is rather modest

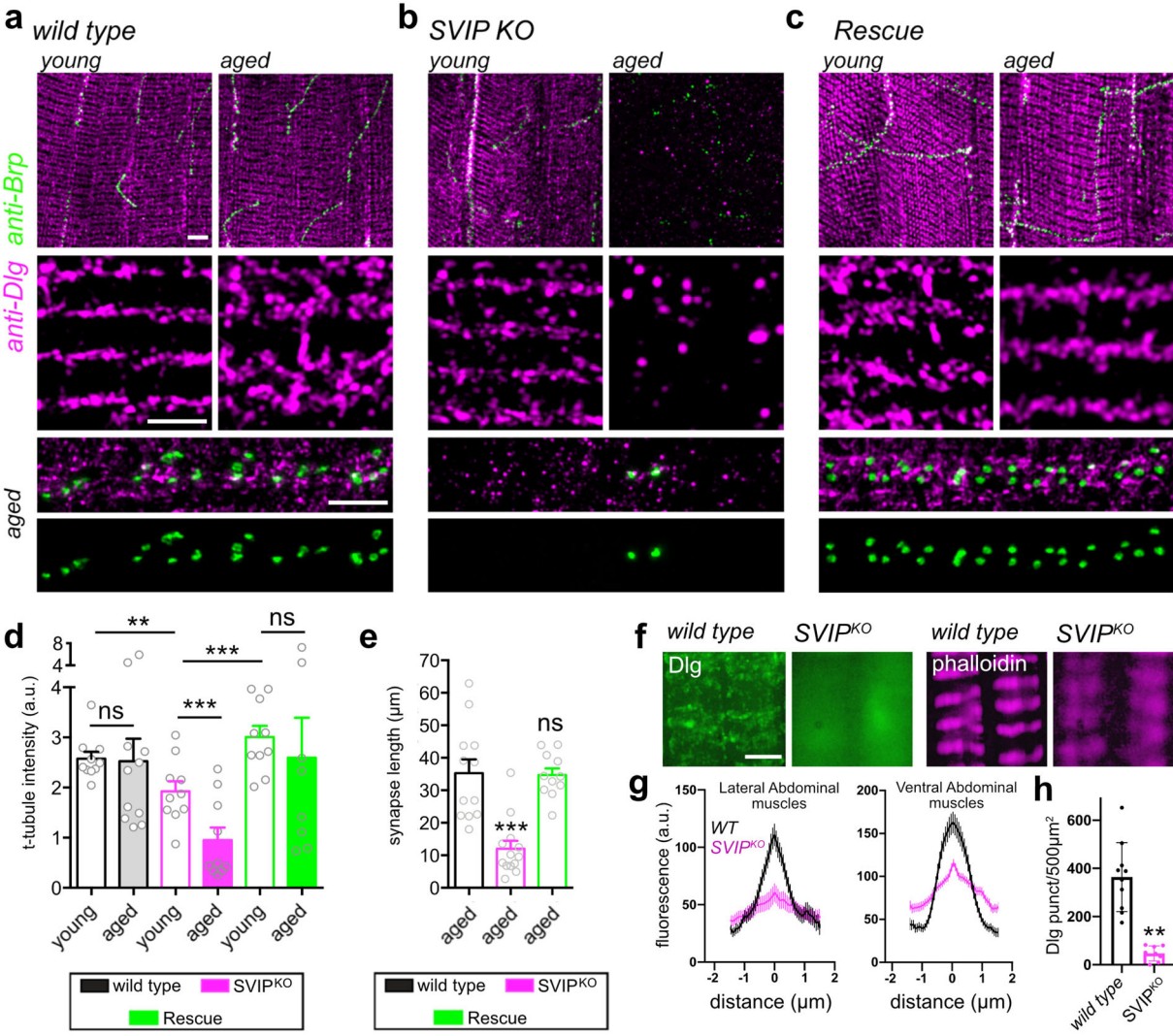

**Fig. 4 NMJ and muscle degeneration in *SVIP^KO*. a–c** Representative images of TTM muscles stained with the pre-synaptic marker, Brp, and post-synaptic marker DLG (labels post-synaptic membranes and T-tubules). Young animals are 1–5 days post eclosion. Aged animals are 25–30 post eclosion. Insets: middle row (square boxes) show anti-DLG stained t-tubules. Bottom rows (rectangle) show the NMJ with an anti-Brp diagnostic of individual active zones. Scale bar 5 μm. **d** Quantitation of T-tubule (DLG) fluorescence intensity. (*WT* young $n = 10$ cells examined over six independent animals, *WT* aged $n = 11$ cells examined over six independent animals, *SVIP^KO* young $n = 10$ cells examined over six independent animals, *SVIP^KO* aged $n = 10$ cells examined over six independent animals, rescue young $n = 10$ cells examined over six independent animals, rescue aged $n = 10$ cells examined over six independent animals). **e** Quantitation of synapse length. (*WT* aged $n = 12$ cells examined over six independent animals, *SVIP^KO* aged $n = 13$ cells examined over six independent animals, rescue aged $n = 11$ cells examined over six independent animals). **f** Representative images of abdominal muscles stained with DLG and Phalloidin. Scale bar 5 μm. **g** Linescan of fluorescence intensity within single phalloidin bands (average ± SEM for each point), data aligned to peak intensity within a band. **h** Quantification of DLG puncta per 500 μm². Data presented as mean and SEM. Student's *t*-test for individual comparisons (*$p < 0.05$; **$p < 0.01$; ***$p < 0.001$). ANOVA for multiple comparisons (*$p < 0.05$; **$p < 0.01$; ***$p < 0.001$).

**Age-related muscle defects in *SVIP* knockout animals.** IBMPFD is an age-related disorder, with patients typically presenting with muscle weakness in their fifth to sixth decade. Thus, we sought to characterize the progression of muscle wasting and neuromuscular degeneration, over time, in adult *Drosophila*. We assessed muscle integrity at 1 week and 4 weeks of age, comparing wild type and *SVIP^KO*. The Dlg antibody labels the highly organized junction of T-tubule membranes with the muscle plasma membrane[32]. We quantified anti-Dlg signal intensity/area (see "Methods" section). The anti-Brp antibody labels pre-synaptic T-bars that reside at sites of neurotransmitter release within the neuromuscular junction. Loss of Brp is an early hallmark of neuromuscular degeneration in *Drosophila*[33].

In wild type, there is no change in Dlg over time (Fig. 4a, d), although minor changes in Dlg organization occur (Fig. 4a, inset).

By contrast, major changes are observed in the *SVIP^KO*. Dlg staining is slightly diminished compared to wild type at 1 week (Fig. 4b, d), but is nearly abolished by 4 weeks (Fig. 4b, d). In parallel, we observe that Brp puncta are dramatically reduced in number (Fig. 4b, bottom inset). We used Brp puncta as a proxy to calculate the length of the neuromuscular synapse and find a highly significant, age-dependent decrease in the *SVIP^KO* (Fig. 4e). Importantly, both the change in Dlg and Brp are fully rescued by muscle-specific resupply of *SVIP* in *SVIP^KO* (Fig. 4c–e). We conclude that loss of *SVIP* causes progressive degeneration of muscle membrane systems and neuromuscular synapses.

In order to extend our anatomical analyses, we used phalloidin as a marker of normal muscle actin organization. These analyses were performed in the adult abdominal muscles. As in the jump muscle (Fig. 4a–e), Dlg labels the t-tubule network, although it

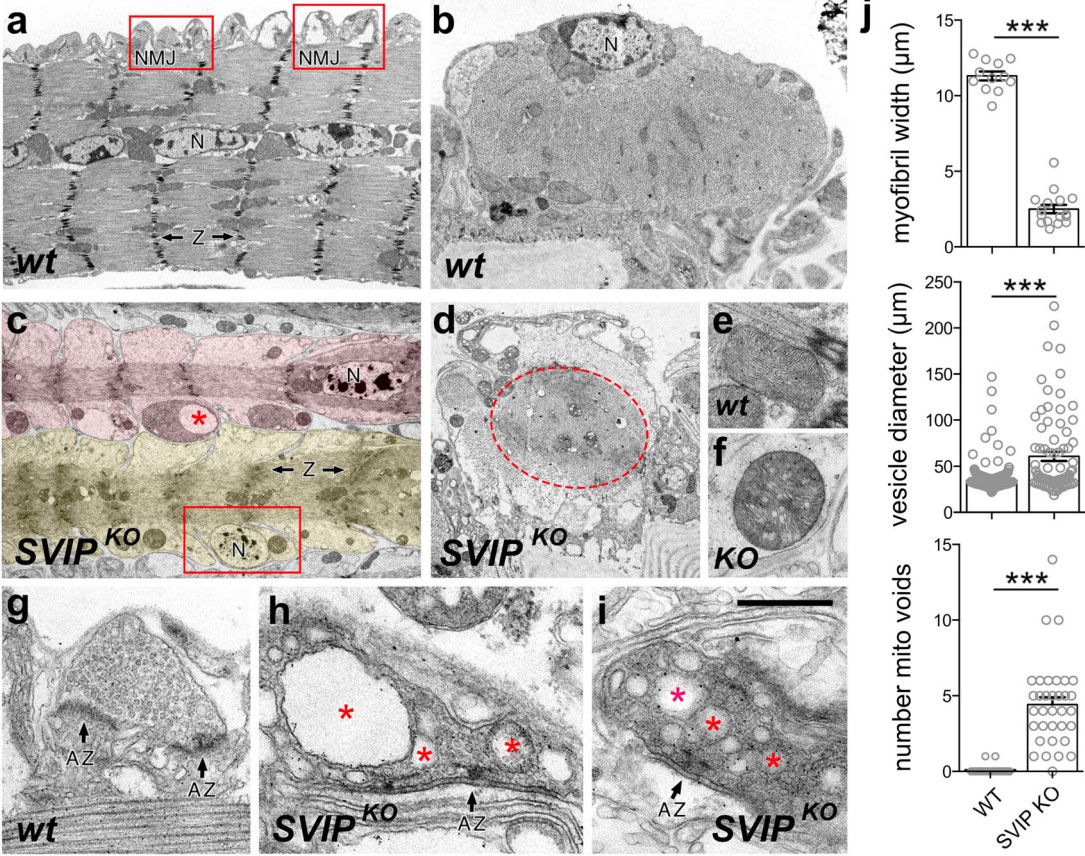

**Fig. 5 Ultrastructural analyses of neuromuscular degeneration in *SVIP^{KO}*. a** Longitudinal section of wild-type (*wt*) adult abdominal muscle with NMJ indicated (red box), muscle nuclei (N), and z-bands (Z). The view encompasses a single muscle fiber. **b** Cross-section of wild-type (*wt*) adult abdominal muscle. **c** Longitudinal section of *SVIP^{KO}* adult abdominal muscle. Two muscle cells exist in the field of view shaded pink and yellow. Altered mitochondria indicated (red asterisk). Peripheral nucleus (N) red box is indicated. **d** Cross-section of *SVIP^{KO}* adult abdominal muscle with a red dotted line indicating the outline of the muscle cell. **e** Representative mitochondria in wild type. **f** Representative mitochondria in *SVIP^{KO}* (KO). **g** NMJ in wild type with active zones indicated (arrows). **h**, **i** NMJ in the *SVIP^{KO}*. Arrows as in **g**. Red asterisks highlight large vacuoles within the pre-synaptic nerve terminal. **j** Quantitation of myofiber width, vesicle diameter, and a number of mitochondrial voids. Scale bar represents **a**, **c** 5.0 μm, **b** 3.1 μm, **d** 4.9 μm, **e**, **f** 1.1 μm, **g** 0.68 μm, **h** 0.45 μm, **i** 0.4 μm. Data presented as mean and SEM. Student's *t*-test for individual comparisons (***$p < 0.001$). See also Supplementary Fig. S5.

appears less striated in these muscles (Fig. 4f). In *SVIP^{KO}* animals, Dlg becomes completely diffuse, consistent with disruption of the t-tubule network (Fig. 4f, h). In addition, the phalloidin-labeled, actin-rich bands are broader and less well-defined than the *SVIP^{KO}* (Fig. 4f, g). These data are consistent with the analyses of flight muscle and became the substrate for a more detailed analysis by electron microscopy (see next section).

**Ultrastructural visualization of muscle wasting in *SVIP* knockouts.** Muscle wasting, quantified by immunohistochemical techniques, correlates with impaired organismal motility, arguing for progressive neuromuscular degeneration. To assess neuromuscular degeneration in greater detail, we examined neuromuscular ultrastructure in adult *Drosophila*. In Fig. 5a, b, we present longitudinal (Fig. 5a) and cross-sections (Fig. 5b) of a single adult wild type, poly-nucleate muscle cell. Nuclei (N) are present in the central region of the muscle cell (Fig. 5a), occupying a peripheral location in cross-section (Fig. 5b). Prominent, well-organized Z-bands are apparent (Z) with associated mitochondria. The neuromuscular synapses (Fig. 5a, red boxes, NMJ) are distributed along the muscle surface. Higher magnification of the NMJ (G) reveals concentrations of synaptic vesicles at electron densities with characteristic T-bar structures that are diagnostic of sites of neurotransmitter release termed active zones (AZ in Fig. 5g).

The *SVIP^{KO}* mutants show signs of severe wasting, impaired mitochondria morphology, and hallmarks of neuromuscular synapse degeneration. The diameter of individual muscle cells is dramatically reduced (Fig. 5c, d, j). Note that two muscle cells are shown (differentially colored in Fig. 5c) in the same physical space that is normally occupied by a single wild-type muscle cell (Fig. 5a). In *SVIP^{KO}* mutants the acto-myosin cytoskeleton is greatly diminished with disorganized Z-bands (Z). The muscle nuclei (N) are also variably positioned, consistent with impaired muscle cell health. Muscle mitochondria in *SVIP^{KO}* mutants are altered with apparent vacuoles (Fig. 5c, red star). More commonly, however, mitochondria in the *SVIP^{KO}* mutants show numerous, smaller voids within the intra-organellar membranes of unknown consequence (compare Fig. 5e, f; also see panel j and Supplementary Fig. S5B, C). The mitochondria phenotypes we observe are recapitulated by muscle-specific over-expression of *UAS-SVIP^{R24,25A}* (Supplementary Fig. S5D) emphasizing that this phenotype is likely related to the loss of SVIP-dependent VCP recruitment to the lysosomal membrane system. We also note that defects in mitochondria are rescued by muscle-specific over-expression of *UAS-SVIP* in *SVIP^{KO}* (Supplementary Fig. S5A). Since the primary effect of SVIP muscle over-expression is the potent re-localization of VCP to lysosomal membranes, we propose that the observed mitochondria defects are secondary to the

integrity of the lysosomal system, a possibility that is consistent with a role for VCP in mitophagy[7].

Finally, neuromuscular synapses show ultrastructural hallmarks of degeneration, consistent with immunohistochemical observations (Fig. 4). Neuromuscular junctions (NMJ) are positioned at the muscle surface in cross-section in both wild type and in *SVIP*[KO] mutants (Fig. 5a). In *SVIP*[KO] mutants NMJ are characterized by a darkened appearance and the proliferation of large vesicles (asterisks H, I; see also panel J for quantification) that are consistent with the appearance of enlarged endolysosomes and increased membrane internalization in other models of neuromuscular degeneration, both in *Drosophila*[34] and mammalian systems[35].

**Muscle *SVIP* expression rescues neuronal degeneration in *SVIP*[KO] mutants.** In mammalian systems, ALS and related neuromuscular degenerative disorders are characterized by synaptic degeneration, muscle wasting as well as degeneration of lower and upper motoneurons[36]. Thus, we tested whether degeneration is restricted to the NMJ, or whether it extends to motoneurons that participate in a simple motor circuit in the adult *Drosophila* central nervous system. We quantified neurodegeneration with single-cell resolution by using *ShakB-Gal4* to drive expression of *UAS-CD8-GFP* in the tergotrochanteral motor neuron (TTMn)[37] (Fig. 6a). The TTMn neuron is born and matures morphologically during pupal development (Supplementary Fig. S6A). Therefore, we quantified neuronal morphology during pupal development as well as during adult lifespan, comparing wild type to *SVIP*[KO] animals. Pupal development was staged as a percentage of total developmental time[37]. At 70% pupa development, the TTMn neurons are fully developed and there is no evidence of degeneration in *SVIP*[KO] animals (Supplementary Fig. S6B, C). However, in *SVIP*[KO] animals TTMn neurons rapidly begin to degenerate beginning at 1 day of adulthood and degeneration becomes progressively more severe over the first 30 days of adult life (Fig. 6b–e). By 30 days, 100% of *SVIP*[KO] animals showed severe signs of motoneuron degeneration, including complete loss of dendrites and somas (Fig. 6b–e). In contrast, wild-type animals exhibited little to no signs of dendrite degeneration by 30 days of adulthood (Fig. 6b–e).

As an independent assessment of neurodegeneration, we assessed the function of a simple, well-defined neural circuit termed the giant fiber (GF) circuit (Fig. 6a) that impinges on the TTMn motoneuron[38]. Electrical stimulation of the GF neuron in the *Drosophila* head elicits activation of both the TTMn motoneuron (Fig. 6a, green) and the DLMn motoneuron (Fig. 6a, blue). It is possible to assess the integrity of this simple circuit by stimulating the GF and recording in the target musculature[37] (Fig. 6a). Muscle responses are categorized as a failure if there is a complete absence of stimulus-time-locked muscle depolarization. Failures will occur if the neuromuscular synapse has completely uncoupled from the muscle, or if elements of the neural circuitry in the central nervous system have degenerated and broken the circuit. Complete synapse elimination at the NMJ is rarely observed (Fig. 4b, d, e), suggesting that complete failures likely reflect degeneration of elements within the central nervous system. Wild-type animals show a 100% success rate at 30 days in both target muscles (Fig. 6f, g). Young *SVIP*[KO] animals (2–5 days) have a 100% success rate, but by 30 days this has decreased to 0% success in the TTMn target muscle and to less than 40% success in the DLMn target muscle (Fig. 6f, g). These data are consistent with our anatomical analyses demonstrating severe motoneuron dendrite degeneration (Fig. 6b–e).

We next asked where *SVIP* is necessary to support the integrity of motoneurons in the CNS. Remarkably, we find that muscle-specific expression of *UAS-SVIP* completely rescues neurotransmission at the TTMn target muscle and doubles the success rate at the DLMn target muscle in *SVIP*[KO] animals (Fig. 6f, g). These results demonstrate that muscle-specific expression of *UAS-SVIP* is sufficient to rescue muscle health, NMJ integrity and it is also sufficient to improve motoneuron integrity. Thus, we suggest that SVIP-mediated recruitment of VCP to lysosomes in muscle can have profound cell non-autonomous consequences for the integrity of central neuronal circuitry and animal behavior. These data are consistent with a growing body of evidence that skeletal muscle proteostasis can have organism-wide effects on cellular health and organismal fitness[39,40] (see "Discussion" section).

The ability of muscle SVIP expression to rescue motoneuron degeneration prompted us to determine the breadth of this effect. To do so, we quantified cell death in aged wild type and *SVIP*[KO] animals, as well as *SVIP*[KO] animals with muscle-specific *SVIP* rescue. Cell death was quantified by counting neuronal soma positively stained by ApopTag (see "Methods" section). In the region of the central nervous system where the majority of motoneurons reside that innervate the limbs, abdominal muscles, and flight muscles (termed ventral nerve cord; shown in Fig. 6) aged *SVIP*[KO] animals show a significant increase in cell death that is fully rescued by muscle-specific expression of *SVIP* (Fig. 6h, i). By contrast, although significantly elevated cell death was observed within the neural circuitry of the head and the visual system in the aged *SVIP*[KO], this cell death was not rescued by muscle-specific expression of *SVIP* (Fig. 6j, k). These data agree with the muscle-specific, cell non-autonomous rescue of motoneurons at the single-cell level (Fig. 6b–g). These data also suggest that the restoration of muscle health by muscle-specific *SVIP* expression in *SVIP*[KO] is achieved by localized trophic support of motor circuitry rather than an organism-wide effect.

**Identification of a *VCP* disease-associated mutation that abolishes VCP–SVIP binding.** To this point, we have focused on the necessity of the *SVIP* gene. We next sought to connect SVIP activity to human disease-relevant mutations in the *VCP* gene. We hypothesize that pathogenic mutations in *VCP* could, in some instances, directly impair the VCP–SVIP interaction, precluding SVIP-dependent VCP recruitment to lysosomes, causing multisystem failure. To test this hypothesis, we biochemically screened 10 conserved, disease-relevant, missense mutations in VCP (Supplementary Table 1) for loss of SVIP binding. Four mutations reside in the N-terminal protein–protein interaction domain, four reside in the D1 ATPase domain and two reside in the first linker region (Fig. 7a). Mutations in the N-terminal region had the largest effect of reducing SVIP–VCP affinity (Fig. 7b–e). The most dramatic effect was observed in the P134L mutant, which reduced SVIP–VCP affinity more than 30-fold (Fig. 7b, c).

Given the marked reduction in SVIP–VCP binding caused by the *VCP*[P134L] mutation, we sought to determine the in vivo consequences of this mutation, comparing them to the effects of the *SVIP*[KO]. To better mimic disease, we used CRISPR to knock-in the *VCP*[P134L] mutation at the endogenous *VCP* locus. An sfGFP tag was fused to the mutant transgene to enable localization studies (VCP[P134L]-*sfGFP*). As a control, we also generated a knock-in of wild-type *VCP* fused to sfGFP (VCP[WT]-*sfGFP*). While we were able to obtain homozygous *VCP*[WT]-*sfGFP* adult flies, we failed to obtain any homozygous *VCP*[P134L]-*sfGFP* adults, suggesting that the mutation is homozygous lethal. Given that IBMPFD is autosomal dominant, *VCP*[P134L] heterozygotes should genetically mimic the human disease and we further characterized the defects associated with the heterozygous mutant.

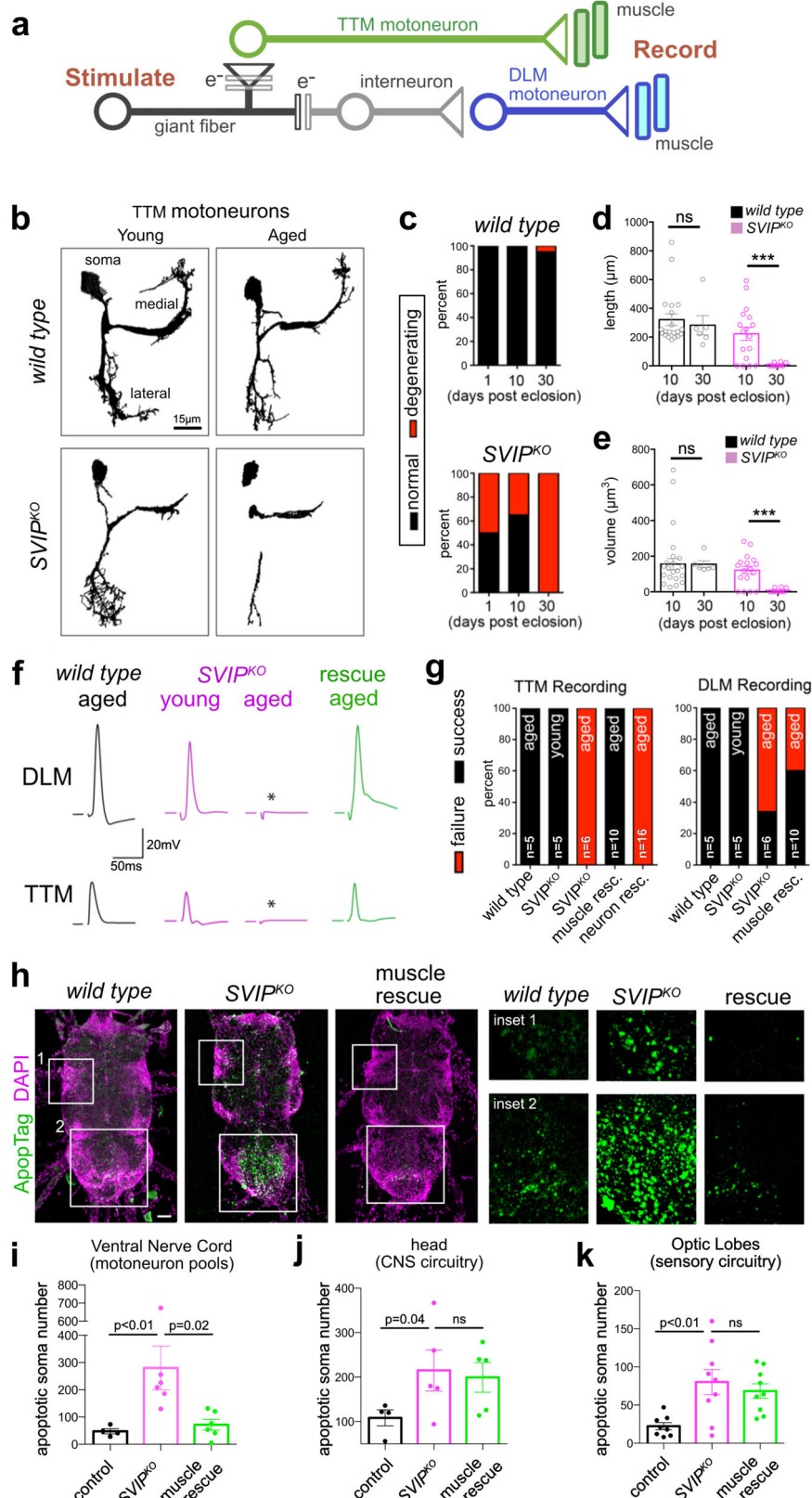

VCP protein self-oligomerizes into a hexamer. Since we are analyzing *VCP^{P134L}* heterozygous mutants, we wanted to examine whether the P134L mutant protein could assemble with wild-type VCP normally in vivo. To assess this, we performed co-immunoprecipitations. The epitope-tagged VCP^{P134L}-sfGFP allows easy resolution of the mutant protein compared to wild-

type VCP by gel electrophoresis. VCP^{WT}-sfGFP heterozygous flies were used to control for the sfGFP tag interfering with VCP oligomerization. Both VCP^{WT}-sfGFP and VCP^{P134L}-sfGFP were able to co-immunoprecipitate wild-type VCP (Fig. 7f), indicating that the P134L mutation does not prevent VCP self-oligomerization. Since VCP hexamers should consist of both

**Fig. 6 Neurodegeneration in the *SVIP^KO* and cell non-autonomous muscle rescue. a** Diagram of the giant fiber neural circuit inclusive of the giant fiber residing in the animal head (gf) the PSI interneuron (int, gray), the DLM (blue), and TTM (green) motoneurons and muscles. Synaptic connections are indicated as electrical (e−, parallel lines), chemical (triangle), or mixed (superimposition of triangle and parallel lines). **b** Representative images of *wild type* and *SVIP^KO* TTM motoneurons at 10 days (young) and 30 days (aged). **c** Quantification of the percentage of total visualized motoneurons that showed evidence of degeneration (see main text) (*WT* 10 days $n = 20$ cells examined over 10 independent animals, *SVIP^KO* 10 days $n = 17$ cells examined over nine independent animals, *WT* 30 days $n = 6$ cells examined over 5 independent animals, *SVIP^KO* 30 days $n = 9$ cells examined over 9 independent animals). **d**, **e** Quantification of TTM dendrite morphology including primary dendrite length (**d**), dendrite volume (**e**). **f** Representative recordings from muscles innervated by the DLM motoneuron (top) or TTM motoneuron (bottom). Ages as in **b**. Stimulus artifact removed for clarity. **g** Quantification of NMJ recordings for indicated muscles and genotypes (TTMn: *WT* old $n = 5$ animals, *SVIP^KO* young $n = 5$ animals, *SVIP^KO* old $n = 6$ animals, muscle rescue old $n = 10$ animals, MN rescue $n = 16$ animals; DLM: *WT* old $n = 5$ animals, *SVIP^KO* young $n = 5$ animals, *SVIP^KO* old $n = 6$ animals, muscle rescue old $n = 10$ animals). **h** Apoptag (green) and DAPI (magenta) staining of ventral nerve cords in the genotypes indicated. Insets are shown at right. Scale bar 70 µm. **i–k** Quantification of apoptotic soma in the ventral nerve cord (**i** control $n = 4$ animals, *SVIP^KO* $n = 4$ animals, rescue $n = 6$ animals), head (**j** control $n = 4$ animals, *SVIP^KO* $n = 4$ animals, rescue $n = 6$ animals) (**j**) and optic lobe (**k**, subset of the head; control $n = 8$ optic lobes examined over four independent animals, *SVIP^KO* $n = 9$ optic lobes examined over five independent animals, rescue $n = 9$ optic lobes examined over 5 independent animals). See also Supplementary Fig. S6. Data presented as mean and SEM. Student's *t*-test for individual comparisons (*$p < 0.05$; **$p < 0.01$; ***$p < 0.001$). ANOVA for multiple comparisons (*$p < 0.05$; **$p < 0.01$; ***$p < 0.001$).

wild-type and mutant VCP protein, we tested whether VCP mutant heterocomplexes displayed reduced binding to SVIP in vivo. Indeed, the VCP heterocomplex immunoprecipitated significantly less SVIP compared to a wild-type VCP complex (Fig. 7g; $p < 0.05$). Given this result, we examined lysosomes in the $VCP^{P134L}$ mutant animals by imaging Spin-RFP. Similar to what we observed in *SVIP^KO* animals, the dynamic lysosomal network is significantly disrupted (Fig. 7h). We document significant defects in tubule length, tubule density, and tubule complexity (junctions per tubule) (Fig. 7h). Moreover, we note significantly less localization of $VCP^{P134L}$-sfGFP at lysosomes (Supplementary Fig. S7A). Thus, a disease-causing mutation in VCP (P134L) impairs both SVIP binding and the lysosomal network in muscle.

**$VCP^{P134L}$ mutant protein aggregates in an age-dependent manner and causes degeneration.** We find that endogenously expressed VCP localizes to tubular structures around the nucleus and in the cytoplasm (Fig. 8a). In young adult muscles (day 1), $VCP^{P134L}$-sfGFP is similarly localized (Fig. 8a). However, in aged animals $VCP^{P134L}$-sfGFP protein forms aggregates that are significantly more abundant than observed for wild-type VCP (Fig. 8a, b). To further explore whether the $VCP^{P134L}$ mutant protein has an increased propensity to aggregate, we performed an in vivo aggregation assay. We found that application of a VCP inhibitor caused $VCP^{P134L}$-sfGFP protein to form aggregates, whereas $VCP^{WT}$-sfGFP did not form aggregates in 3rd instar larval muscles (Supplementary Fig. S7b). It appears that age and cellular stress can trigger the formation of aggregates that contain the $VCP^{P134L}$ mutant protein. It is possible that the $VCP^{P134L}$ mutant protein is aggregation prone or $VCP^{P134L}$ recognizes aggregates that form due to impaired $VCP^{P134L}$ ability to target the lysosome. In either case, our data are consistent with observations in human IBMPFD patients where protein aggregates stained positively for VCP protein[10].

Finally, we examined the organismal defects associated with the $VCP^{P134L}$ mutation (Fig. 8c–f). The lifespan of $VCP^{P134L}$ mutants was modestly, but significantly, reduced compared to $VCP^{WT}$ (Fig. 8e, $p < 0.001$) and displayed progressive mobility defects (Fig. 8f). We then examined neuronal function at the DLM neuromuscular junction in adult *Drosophila*. It is well established that this synapse can reliably evoke muscle depolarization when stimulated at 100 Hz. When we repeated this stimulation paradigm in aged control animals, the motor neuron could evoke a response for every stimulus (Fig. 8c). However, $VCP^{P134L}$ aged mutant animals exhibited a 70% failure rate (Fig. 8c). Direct stimulation of the motoneuron evokes successful muscle

depolarization, demonstrating that failures do not originate at the NMJ (Fig. 8c, d). Rather, this phenotype is consistent with degeneration of central processes, such as the motoneuron dendrite degeneration in the *SVIP^KO* as well as widespread cell death in the *SVIP^KO* (Fig. 6). Finally, we quantified NMJ integrity based on the release site quantification (anti-Brp staining). We find a significant reduction in Brp puncta in the $VCP^{P134L}$ aged mutant compared to control and numerous boutons that completely lack Brp puncta (Fig. 8g, h), indicative of synapse degeneration[33]. Note, however, that there was limited evidence of complete synapse elimination, consistent with ongoing synapse degeneration and continued neurotransmission from nerve to the muscle. Taken together, the organismal and cellular defects observed in $VCP^{P134L}$ mirror many of the phenotypes in the *SVIP^KO*.

**Identification of a human *SVIP* mutation.** In order to explore additional connections between *SVIP* and human disease, we screened a cohort of sporadic FTD patients ($n = 341$), with whole-genome sequencing data from the UCSF Memory and Aging Center (MAC), for putative pathogenic variants in *SVIP*. We found one potentially deleterious heterozygous nonsynonymous single nucleotide variant (SNV) in *SVIP* at genomic coordinates Chr11:22,844,669. This SNV results in a C to T substitution at nucleotide 230 in the coding sequence (NM_148893: c.230 C>T) and was subsequently confirmed with Sanger sequencing. This variant causes a serine to leucine substitution, p.S77L, with the substituted amino acid being highly conserved from *Drosophila* to human (refer back to Fig. 1b). Importantly, we did not find the c.230 C>T or any other nonsynonymous *SVIP* variants in 106 clinically normal, healthy older controls or 318 early-onset Alzheimer's disease patients from the UCSF MAC cohorts. Further, to the best of our knowledge, this variant has not been reported in public whole-genome and/or exome sequencing databases, including the Genome Aggregation Database (gnomAD) and Exome Aggregation Consortium (ExAC), suggesting that this variant is extremely rare in the general population. Finally, this patient was negative for known pathogenic mutations in the following genes: *APP*, *C9ORF72*, *CHMP2B*, *FUS*, *GRN*, *MAPT*, *PSEN1*, *PSEN2*, and *TARDBP*.

**$SVIP^{S82L}$ disrupts lysosome structure and function in *Drosophila*.** To examine potential defects associated with the human S77L mutation, we first examined SVIP–VCP binding. We generated a GST fusion protein of *Drosophila* SVIP harboring the orthologous mutation (S82L, see Fig. 1b for reference) and found

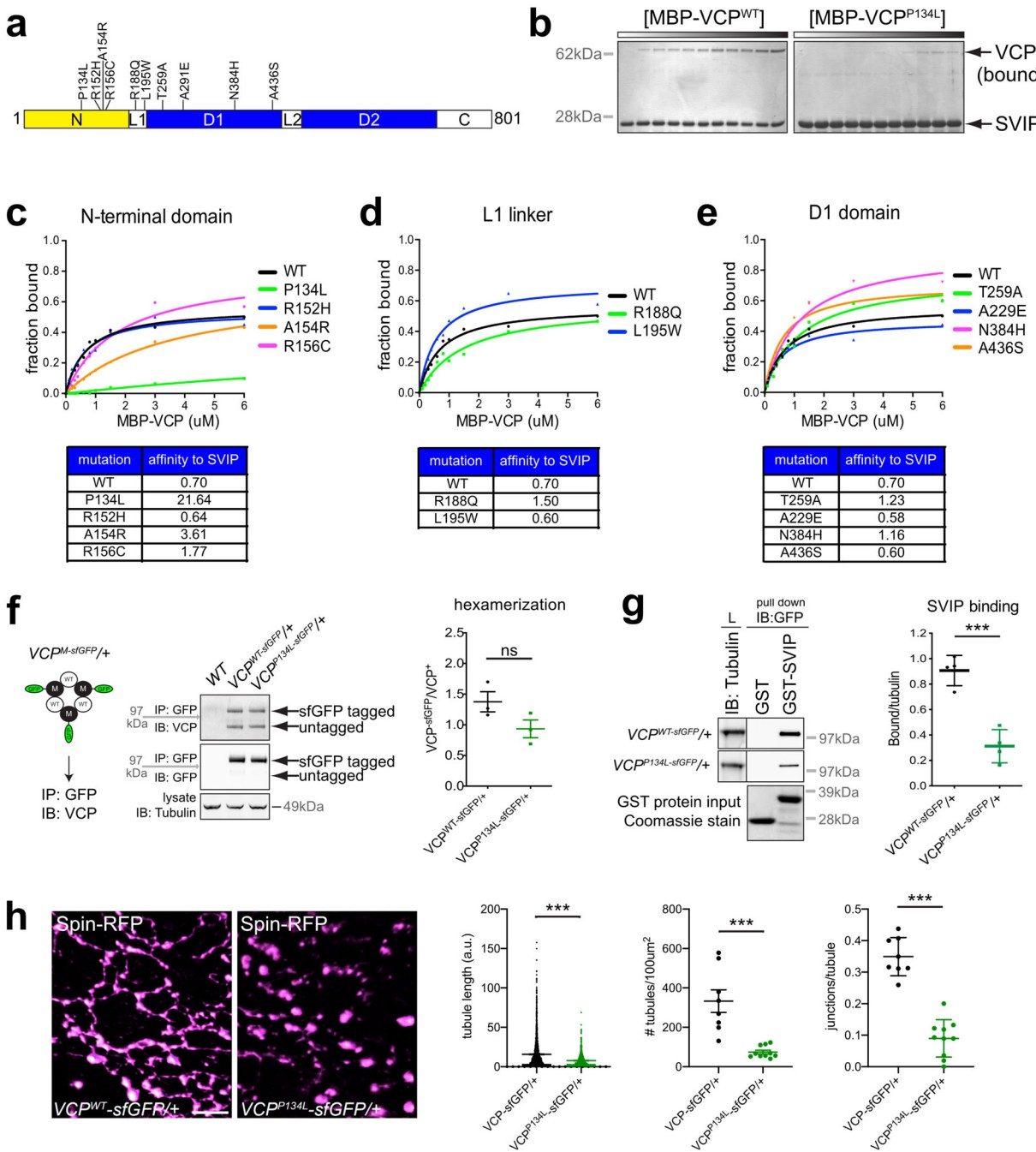

**Fig. 7 VCP disease mutation abrogates SVIP-VCP interaction. a** Schematic of VCP with relative positions of disease-relevant mutations. **b** In vitro binding titration assay of GST-SVIP with MBP-VCP^WT and MBP-VCP^P134L recombinant proteins. **c–e** Binding curves of GST-SVIP with each MBP-VCP mutant as indicated. Affinities are listed in the tables for each mutant. **f** Co-immunoprecipitation of endogenously expressed VCP and VCP-sfGFP (transheterozygote). Diagram of hexameric VCP protein with proposed SVIP (green) binding at left. Quantification at right. (*n* = 3 independent experiments). **g** In vitro binding assay with recombinant GST-SVIP and VCP-sfGFP protein lysates extracted from whole flies. Quantitation of the amount of VCP-sfGFP protein bound to GST-SVIP. (*n* = 3 independent experiments). **h** Representative images of Spin-RFP in *VCP^wt* and *VCP^P134L* mutants. Quantification at right. (*n* = 8 muscle cells examined over four independent animals). Data presented as mean and SEM. Student's *t*-test for individual comparisons (**p* < 0.05; ***p* < 0.01; ****p* < 0.001).

that SVIP^S82L can still bind VCP in vitro (Supplementary Fig. S8). Next, we generated a *Drosophila SVIP^S82L* transgene. When *SVIP^S82L* is over-expressed in an *SVIP* heterozygous knockout (*SVIP^KO*/+) background we find that lysosomal tubules are no longer visible and lysosomal dynamics are severely perturbed (Fig. 9a; Supplementary Movies 1 and 2). Finally, an *SVIP^S82L* knock-in, when tested in a lifespan assay, also showed resistance to the effects of tunicamycin feeding, similar to *SVIP^KO* mutants

(Fig. 9b). Thus, although SVIP^S82L can still bind VCP in vitro, the mutant protein is sufficient to dominantly disrupt tubular lysosomes and affect animal physiology.

Finally, we imaged the mCh-GFP-Atg8a autophagy sensor to compare the proficiency of autophagosome–lysosome fusion in animals over-expressing *SVIP^WT* or *SVIP^S82L* in the *SVIP* heterozygous knockout background (*SVIP^KO*/+) (Fig. 9c, d). When *SVIP^WT* is over-expressed, we observed little to no GFP

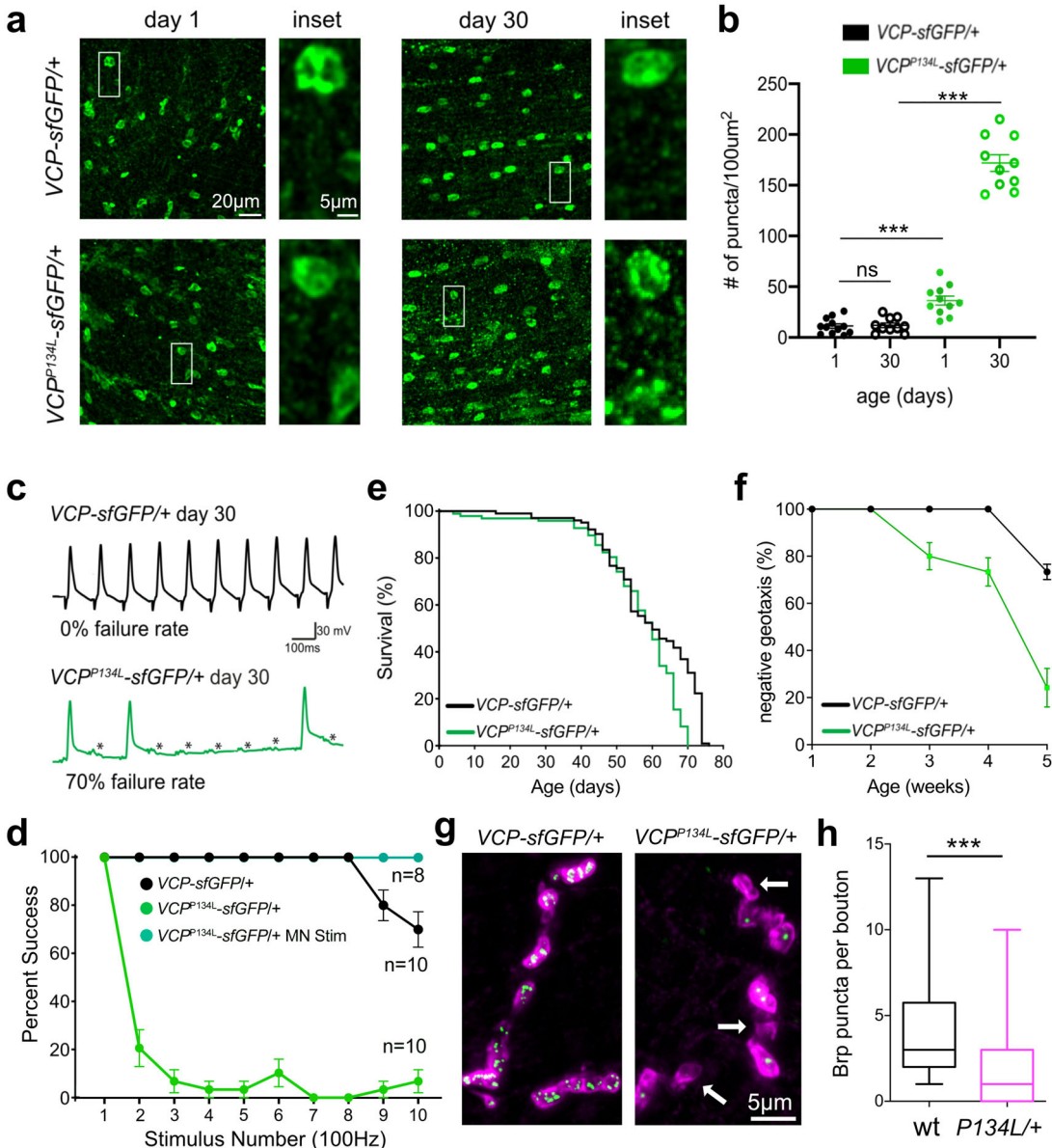

**Fig. 8 $VCP^{P134L}$ knock-in mutants cause VCP aggregation and functional degeneration. a** Localization of endogenously expressed VCP$^{WT}$-sfGFP and VCP$^{P134L}$-sfGFP in young and aged TTM muscles. **b** Quantification of puncta density for data in **a**. Data presented as mean and SEM. **c** Representative traces recorded from the DLM muscle in wild type and $VCP^{P134L}$. Failures are indicated by asterisks. Average failure rates are indicated below each trace ($n = 10$ animals for each genotype), stimulus artifact removed for clarity. **d** Quantification of data in **c** for the genotypes indicated. Stimulation was achieved either through activation of the giant fiber (**a**) or direct motoneuron stimulation (MN stim). **e** Lifespan of VCP$^{WT}$-sfGFP ($n = 103$) and VCP$^{P134L}$-sfGFP ($n = 97$). **f** Negative geotaxis assay for genotypes at indicated ages. Data from three independent experiments with 20 animals each are presented as mean and SEM for each age point. **g** Representative images of NMJs stained with Brp (green) and DLG (magenta). **h** Quantification of Brp per bouton. Data presented in a box and whisker plot (minimum, 1st quartile, median, 3rd quartile, and maximum). Scale bars indicated on images. See also Supplementary Fig. S7. ANOVA for multiple comparisons (*$p < 0.05$; **$p < 0.01$; ***$p < 0.001$).

fluorescence where mCherry fluorescence was observed, indicating that autophagosomes are efficiently fusing with lysosomes. However, when $mCh$-$GFP$-$Atg8a$ is co-expressed with $SVIP^{S82L}$, we observed a significant increase in GFP fluorescence, indicating autophagosome–lysosome fusion impairment (Fig. 9c, d). Collectively, these data suggest that the $SVIP^{(S82L/S77L)}$ mutation is pathogenic in *Drosophila* muscle.

## Discussion

IBMPFD is characterized by degeneration of tissues, including the neuromuscular junction and central neurons, causing similar symptoms associated with ALS[3,4], Parkinson's[7,8], and FTD[9]. Although IBMPFD is a comparatively rare disease, it has the potential to inform about generalized mechanisms that drive neurodegeneration more broadly. Here, we define SVIP as a cytoplasmic co-factor that is responsible for the translocation of VCP to a dynamic, tubular-lysosomal system in muscle. The importance of SVIP is underscored by our phenotypic analysis of *SVIP* knockouts and *VCP* transgenic knock-in mutants that disrupt the SVIP–VCP interaction. The $SVIP^{KO}$ shows a progressive, degenerative phenotype in muscle and motoneurons that is similar to that observed in disease-relevant *VCP* mutants as well as patients with IBMPFD and ALS. Degeneration is associated

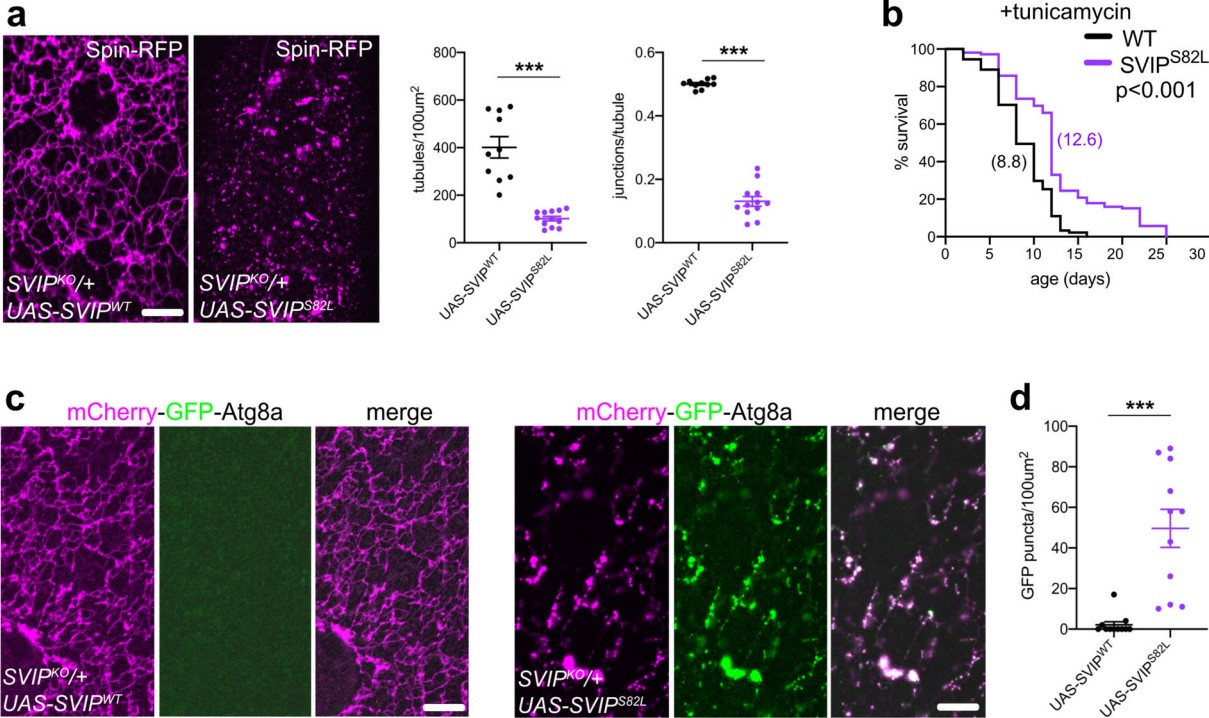

**Fig. 9 SVIP variant that disrupts lysosome integrity and function. a** Representative images of Spin-RFP (lysosomes) in 3rd instar larval *Drosophila* muscles for the genotypes indicated. At right, quantification of tubule density (tubules/100 $\mu m^2$) and junctions per tubule. ($n = 10$ muscle cells examined over five independent animals). **b** Lifespan of *WT* ($n = 91$) and *SVIP$^{S82L}$* ($n = 106$) flies fed tunicamycin ($p < 0.0001$). **c, e** Representative images of mCherry-GFP-Atg8a (autophagy sensor) in 3rd instar larval *Drosophila* muscles for the genotypes indicated. **d** Quantification of data from **c**. ($n = 10$ muscle cells examined over five independent animals). Scale bars, 5 $\mu m$. See also Supplementary Fig. S8 and Supplementary Movies S1 and S2. Data presented as mean and SEM. Student's *t*-test for individual comparisons (***$p < 0.001$).

with the disintegration of intracellular membrane systems and disrupted mitochondria. We show that these phenotypes extend to progressive deficits in organismal behavior and diminished lifespan.

**Regulated deployment of VCP by SVIP and related co-factors.** Our data speak to the possibility that VCP is deployed to different organelles based upon stress-related or cell-specific needs (Fig. 1a). This is consistent with the prior demonstration that VCP is recruited to mitochondria following pharmacological poisoning of oxidative phosphorylation with cyanide *m*-chlor-ophenyl-hydrazone (CCCP)[7]. Given the diverse activities of VCP, it was initially surprising that targeting the majority of VCP to lysosomes in muscle could have a beneficial outcome for orga-nismal lifespan. Different cell types may routinely confront dif-ferent stressors and emphasize different VCP co-factors. The fact that muscle lysosomes exist as an expansive, dynamic tubular network implies the importance of lysosomes to muscle biology and could underscore why *SVIP* knockouts have a particularly profound effect on muscle, with ensuing ALS-like degeneration of the neuromuscular junction and motoneuron dendrites.

**SVIP controls organismal health and lifespan under conditions of lysosomal stress.** Our data suggest that SVIP is a context-specific survival factor. We demonstrate that *SVIP* over-expression, which is sufficient to drive VCP to lysosomes, is also sufficient to enlarge the dynamic tubular-lysosomal network in muscle. Muscle-specific *SVIP* over-expression doubles lifespan when organisms are reared on food containing a lysosomal stressor, BafA. In one sce-nario, the loss of lysosomal pH caused by BafA is counteracted by the expansion of the tubular-lysosomal system, caused by *SVIP*

over-expression. However, it is possible that the size and extent of the lysosomal system are less relevant compared to the increased abundance of VCP at the lysosomal compartment. In either case, SVIP-dependent recruitment of VCP to muscle lysosomes drama-tically extends lifespan under conditions of lysosomal stress. Future efforts might be directed toward assessing whether *SVIP* over-expression can ameliorate the pathology and etiology of diseases associated with impaired lysosomal integrity. For example, lysoso-mal storage diseases are a group of ~50 rare metabolic disorders that primarily affect children, causing lethality within months or a few years of age.

It is important to emphasize that, while blocking SVIP-mediated VCP localization to lysosomes (*SVIP* knockout) is maladaptive if animals are fed bafilomycin-A, it is beneficial if animals are fed tunicamycin, stressing the ERAD pathway. We interpret this effect as being a consequence of more VCP being available to the ER system in the absence of lysosomal localization, consistent with the initial identification of SVIP as a negative regulator of ERAD[24]. Thus, SVIP can be considered a context-specific survival factor. This finding is also consistent with our model regarding the dynamic deployment of a limited VCP resource to sites of cellular stress (Fig. 1a).

It is also worth noting that reduced proteostasis is a general feature of muscle aging and degeneration[41]. Indeed, proteostasis mechanisms naturally decline with age[42] and are thought to be a major contributing factor to the onset of neurodegenerative diseases[43–45]. The predominant cellular degradation sites for clearing and recycling cellular waste products are lysosomes. Enhancing autophagy–lysosome-mediated proteostasis either pharmacologically or by caloric restriction has been shown to increase organismal healthspan in multiple species, including worms, flies, and mammals[46].

The sites of action for the beneficial effects of caloric restriction and enhanced autophagy–lysosome mediated proteostasis are under investigation. Intriguingly, there have been multiple examples of muscle proteostasis influencing the broader health of an organism. In particular, disrupting proteostasis in *C. elegans* muscles causes upregulation of the heat shock response system in distant tissues via the expression of specific chaperone proteins[39]. In *Drosophila*, the muscle-specific expression of *FOXO* enhances organismal viability[40]. These "tele-proteostasis" mechanisms may promote organismal robustness in a stressful environment by communicating the existence of proteotoxic stress to surrounding, healthy tissue, enabling a preparatory response. In essence, this is a type of feed-forward stress signaling mechanism that promotes organismal robustness.

**Models of IBMPFD.** Current *Drosophila* models of IBMPFD are based upon transgene over-expression, with unknown consequences on the normal regulation of VCP protein throughout the cell. In particular, it has been proposed, based on transgenic over-expression models, that IBMPFD is associated with adverse VCP gain of function activity[22]. By extension, it was proposed that VCP inhibitors might be beneficial to disease progression and human health[22]. However, the complexity of VCP signaling, acting at many sites within the cell (Fig. 1a) seems to argue for other therapeutic approaches.

To facilitate future therapeutic approaches, we have generated several mutants and transgenic animals that can be considered *Drosophila* models for exploration of IBMPFD. The $VCP^{P134L}$ knock-in and the $SVIP^{KO}$ specifically target the SVIP-dependent recruitment of VCP to lysosomes. It is interesting that several pathogenic VCP mutations do not affect SVIP binding. Yet, there is clear phenotypic overlap comparing the loss of VCP[16] to the $VCP^{P134L}$ knock-in and the $SVIP^{KO}$. Future work will be required to determine how the diversity of pathogenic *VCP* mutations are linked to autophagosomal–lysosomal dysfunction. Regardless, our data suggest that targeting the interface of VCP with specific cofactors such as SVIP may allow modulation of VCP activity at specific intracellular sites, with the potential to ameliorate mutation-dependent diseases without compromising VCP activity globally throughout the cell.

Finally, we identify a human patient with FTD harboring a mutation in *SVIP* and demonstrate that this mutation causes pathological defects in *Drosophila* muscle lysosomes and impairs autophagy that parallels the phenotypes of our other *Drosophila* disease models. The fact that *SVIP* is a small gene makes it unlikely that a substantial patient population will harbor *SVIP* mutations, possibly precluding the determination of whether *SVIP* mutations are actually causal for human disease. None-the-less, the fact that the patient-derived mutation is pathogenic in *Drosophila* muscle, phenocopying other disease models, argues that *SVIP* may be relevant to the human condition and encourages further investigation into the potential benefits of *SVIP* over-expression as a potential suppressor of human disease.

## Methods

**Fly stocks and molecular cloning methods.** The following transgenic fly stocks were generated in this study: *SVIP-sfGFP* (endogenous gene replacement), *dVCP-sfGFP* (endogenous gene replacement), $SVIP^{KO}$, $SVIP^{S82L}$, UAS-SVIP, UAS-SVIP-mCherry, UAS-SVIP$^{R24,25A}$-mCherry, UAS-FKBP$^{F37V}$-Spin-mCherry, UAS-FRB-dVCP-Venus, UAS-SVIP$^{S82L}$. *Drosophila VCP, SVIP,* and *Spin* cDNAs were obtained by amplifying from DGRC (*Drosophila* Genomics Resource Center, Bloomington, IN) clones GM02885, GH02734, and AT25382, respectively. The *FKBP$^{F37V}$* and *FRB* constructs were purchased from Addgene. The cDNAs were cloned into the Gateway pENTR vector (ThermoFisher) and subsequently cloned into destination vector pTWV obtained from the *Drosophila* Gateway Vector Collection (Carnegie Institution, Baltimore, MD). Injection services of BestGene were used to generate transgenic flies. *UAS-GFP-mCherry-Atg8a* stock was purchased from the Bloomington Stock Center.

Endogenous gene replacements were generated using a scarless CRISPR strategy (http://flycrispr.molbio.wisc.edu/scarless). In brief, donor and gRNA plasmids were co-injected into *vas-Cas9* expressing *Drosophila* embryos (either BDRC 56552 or 51324 stocks). The donor plasmid contained the transgene of interest, ~1 kb of homologous sequence upstream and downstream of the insertion sites, and a dsRed cassette as a selection marker. Additionally, the PAM sites that were used to guide Cas9 cleavage of the endogenous genome were mutated in the donor plasmid to prevent cleavage of the transgene after insertion. The gRNA plasmid contained two gRNA sequences to direct Cas9 cleavage of sites upstream and downstream of the target genomic region to be replaced. The injection services of Bestgene were used to generate transgenic flies. Transgenic animals were verified by PCR and sequencing and the dsRed cassette was removed by crossing flies to 3xP3 transposase expressing flies. See Supplementary Table 2 for a list of all primers used in this study.

**In vitro binding assays.** MBP and GST fusion proteins were affinity purified in column buffer (20 mM Tris (pH 7.0), 150 mM NaCl, 2 mM EDTA, and 0.1% NP40) with Amylose Resin (NEB) or GST-bind resin (Millipore), respectively. MBP fusion proteins were eluted with maltose and GST fusion proteins were eluted with Glutathione per manufacturers recommendations. For binding reactions, 1 μg of each protein was incubated with 50 μl GST-bind resins in NP40 buffer for 1 hr. Beads were washed 3× with NP40 buffer before 50 μl of 5× Sample Buffer was added. Proteins were boiled for 10 min and resolved by SDS-PAGE on a 4–12% Bis-Tris gel (ThermoFisher). Protein gels were stained with Coomassie Blue (Biorad) for 30 min and destained for 2–3 h in destain buffer (40% MeOH and 10% Acetic Acid) before imaging. For in vivo pull-down assays, purified recombinant proteins were left on beads and incubated with protein lysates in NP40 buffer for 2 h. Beads were washed 3× with NP40 buffer before adding 50 μl of 5× sample buffer. Proteins were separated by SDS-PAGE and immunoblotted with anti-VCP (Cell Signaling, 7F3, Rabbit mAb #2649) and anti-GFP (mouse monoclonal, Invitrogen clone 3E6; A-11120). Total lysates were immunoblotted with anti-tubulin (E7-c, DSHB). HRP conjugated goat anti-mouse and goat anti-rabbit secondary antibodies (ThermoFisher) were used at 1:2000.

**Microscopy methods.** For live imaging, 3rd instar larvae or adult fly abdomens were dissected in HL3 saline buffer with no calcium (70 mM NaCl, 5 mM KCl, 10 mM MgCl$_2$, 10 mM NaHCO$_3$, 115 mM sucrose, 4.2 mM trehalose, and 5 mM HEPES) and all live imaging was performed in saline. Imaging was performed on an inverted Axiovert 200 microscope (Zeiss) using a 100× Plan Apochromat objective (1.4NA). Images were captured with a CoolSnap HQ2 CCD camera (Photometrics) and de-convolved using Slidebook 5.0 software (Intelligent imaging innovations, Denver, CO). Image quantification was performed using ImageJ software (NIH). Volume rendering was performed with Slidebook 5.0 software. Any adjustment of brightness or contrast was performed using Slidebook 5.0 software and always applied to the entire image.

For fixed imaging, adult brains were dissected in O'dowd's saline buffer (101 mM NaCl, 1 mM CaCl$_2$, 4 mM MgCl$_2$, 3 mM KCl, 5 mM glucose, 1.25 mM NaH$_2$PO$_4$, and 20.7 mM NaHCO$_3$, pH 7.2[47]) and fixed in 4% PFA for at least 1 h. Adult abdominal muscles, DLMs, and TTMs were fixed in 4% PFA for 1 h. Muscles were stained with Phalloidin (ThermoFisher) overnight in a final concentration of 0.165 μM. Adult NMJs were visualized using mouse anti-BRP (1:100; Developmental Studies Hybridoma Bank) and rabbit anti-DLG (1:1000[48]). Conjugated secondary antibodies (Alexa Fluor 488 goat anti-mouse and Cy3 goat anti-rabbit) were used at 1:300 (Jackson immuno-research laboratories). The DLMs and TTMs were dissected out of the thorax and identified by three parameters: size, shape, and location in the thorax. For co-localization analyses, Mander's correlation coefficients were calculated using the Coloc2 plugin in FIJI.

**ApopTag staining.** The central nervous system was dissected out of 30-day-old adult *Drosophila*, across indicated genotypes. All fat bodies and trachea were removed from the nervous tissue because it is auto-fluorescent. Tissue was adhered to a glass coverslip with Vectabond (Vector labs) and fixed in 1% PFA overnight at 4 °C. 1% PFA in PBS was prepared fresh, as premixed PFA in a methanol stabilizer disrupts ApopTag assays. Post-fix processing was performed in Ethanol:Acetic acid (2:1) for 10 min at −20 °C. ApopTag assay was performed as described in Millipore Sigma ApopTag Fluorescein kit protocol. Tissue was mounted in vectasheild mounting media (Vector Labs) containing DAPI as a counterstain to image nuclei in the sample. ApopTag Fluorescein signal was imaged in the FITC channel. Samples were imaged on a Nikon Ti-E spinning disc confocal microscope with an Andor Zyla 5.2 camera, using micromanager software. Images were captured with a Plan Apo λ ×20/0.75 Nikon objective. ApopTag Fluorescein positive nuclei were counted using FIJI 3D Object counter.

**EM methods.** Animals were fixed with 2% glutaraldehyde in 0.1 M Nacacodylate buffer, pH 7.3, for 2 h at room temp, rinsed in buffer, and post-fixed with 1% OsO$_4$ in a buffer for 1 h at room temp. The post-fixed samples were then rinsed with water, stained en bloc with 5% uranyl acetate in water, dehydrated in ethanol and propylene oxide, and embedded in Eponate 12 resin (Ted Pella, Inc., Redding, CA). 50 nm sections were cut with a Leica UCT ultramicrotome using a Diatome

diamond knife, picked up on slot grids with Pioloform films, stained with uranyl acetate and Sato's lead, and examined with an FEI T12 TEM at 120 kV equipped with a Gatan Ultrascan 4k × 4k camera.

**Lifespan and climbing assays**. Adult male flies were collected within 12 h of eclosion and maintained at 25 °C on standard fly food. Flies were flipped to new food and scored for death every 2–3 days. Climbing assays were performed by flipping flies into a 100 ml graduated cylinder, tapping them to the bottom, and scoring the number of flies that could successfully climb to the 100 ml mark in 1 min. The statistical software, OASIS[49], was used to calculate the mean lifespans and perform log-rank tests to determine statistical significance.

**Electrophysiology methods**. Adult *Drosophila* was immobilized in dental wax, ventral side down and a glass microelectrode was directly placed into the identified muscle through the cuticle of the animal[37]. Glass electrodes were filled with 3 M KCl (40–60 MΩ resistance) and signals were amplified with an Axopatch 200B (Axon Instruments). Giant fibers were stimulated extracellularly with a Master 8 stimulator through tungsten wire electrodes placed in each of the *Drosophila* eyes. An Axon Instruments Digidata 1322 A Data Acquisition System was used to digitize the data and recordings were collected with Clampex 10.7 software (Molecular Devices). Muscle depolarization at low (0.2 Hz) and high stimulus frequencies (100 Hz) were recorded to test the fidelity of the circuit in control and experimental animals.

**Sequencing data generation and processing**. Whole-genome sequencing of the patient's genomic DNA isolated from whole blood was performed as described previously[50] at HudsonAlpha Institute for Biotechnology (Huntsville, AL) on an Illumina HiSeq-X, with 150 bp paired-end reads to obtain 30× sequencing coverage. Paired-end reads were aligned to the GRCh37 build of the human reference genome with the Burrows-Wheeler Aligner (BWA-MEM v0.7.8) and processed using the Broad Institute's Genome Analysis Toolkit (GATK) best-practices pipeline. The variant was confirmed with direct Sanger sequencing at UCSF.

**Statistical methods**. All Student's *t*-test were performed with two-tailed analysis. All ANOVA are one-way with Tukey's multiple comparisons. Lifespan data were compared using a log-rank test (see Lifespan, above). Analyses performed with Prism v9 (Graphpad).

**Reporting summary**. Further information on research design is available in the Nature Research Reporting Summary linked to this article.

## Data availability

Further information and requests for resources and reagents should be directed to and will be fulfilled by the corresponding author Graeme Davis (Graeme.Davis@ucsf.edu). All unique/stable reagents generated in this study are available without restriction. This study did not generate data sets for public repositories or new data code. Source Data are provided in this paper.

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

## Acknowledgements

This study was funded by NINDS grant R35NS097212 (G.W.D.); NINDS grant K99/R00NS100988 (A.E.J.); NIH-NIA K01 AG049152 (J.S.Y.); NIH-NIA R01 AG062588 (J.S.Y.); Rainwater Charitable Foundation (J.S.Y.); NIA P01 AG019724 (B.L.M.); NIA P50 AG023501 (B.L.M.); and NIA P30 AG062422 (B.L.M.).

## Author contributions

A.E.J. designed and conducted experiments including imaging, biochemistry, genetics, transgene and CRISPR transgenics, lifespan analyses, wrote, and edited the manuscript. B.O.O. contributed anatomical analysis of synapse and dendrite degeneration, cell death, and electrophysiology, and edited manuscript. R.D.F. contributed to electron microscopy and analysis. L.A.P. contributed to image analysis. E.G.G., A.J.M., J.S.Y., and B.L.M. contributed to clinical data. G.W.D. designed experiments, coordinated experiments, secured funding, wrote, and edited manuscript.

## Competing interests

The authors declare no competing interests.
