## [Peer Review File · Nature Communications]

Reviewers' comments:

Reviewer #1 (Remarks to the Author):

The manuscript by Johnson et al explores the effects of SVIP KO, a VCP cofactor, in drosophila. Specifically, they find that SVIP recruits VCP to tubular lysosomes and loss of SVIP phenocopies VCP KO from their previous paper and to some extent a VCP mutation that effects SVIP binding. They then further suggest that SVIP may be involved lysosomal proteostasis and even participate in a non cell autonomous aspect of protection. Finally, they identify a single patient and variant in a patient with sFTD and find that this variant disrupts lysosomal function. Overall, the manuscript and study is interesting as little is understood about SVIP function and VCP co-factor specific functions are only beginning to be teased out. However, the study is extensive leading to several areas that are unconvincing.

- 1) In Figure 1, how do we know for sure that SVIP is on lysosomes. I appreciate that spin is a lysosomal protein but it might be nice to have another marker that confirms lysosomal localization. Figure 1E, wouldn't it be more important to quantitate the amount of VCP on lysosomes (SPIN colocalization) rather than loss from the nucleus.
- 2) Figure 2 demonstrates that VCP can be recruited in an artificial system. Why is VCP in the endogenous state not on the lysosome in control flies. It seems to only be present when SVIP is overexpressed (Figure 1E) or when VCP:SVIP co-localization occurs. Where is basal VCP what recruits it normally to lysosomes?
- 3) It is essential to demonstrate that bafilomycin is truly effecting autophagy at the dose used. More importantly and significant problem is the treatment with tunicamycin. What is the level of ER stress that is present already in the SVIP KO. The claim that is made about SVIP loss leading to enhanced proteostasis via ER stress is not well justified and relies on this experiment alone.
- 4) Figure 4 muscle morphology is intriguing however, it is unclear why more traditional markers are not used focusing on structural elements. DLG is a tubule marker and may be altered with lysosomal pathology as it is within the vesicular/endomembrane system. The EM helps but would like to see a different marker of morphology.
- 5) The focus on a non-cell autonomous aspect of SVIP functionality on degeneration and subsequent rescue is interesting but again, no evaluation of neuronal pathology and relies on a single set of experiments reduces enthusiasm and is again where this manuscript takes a detour from a straightforward story.
- 6) Figure 7. I would like to know what are the comparable human mutations. Is the A154R supposed to correspond the G157R mutation. What is the A291E in humans. Can the biochemistry be quantitated? It is difficult to judge the magnitude of the non P134L mutations. For 7H, where is VCP? is on Spin RFP structures. Or is it just affecting tubulation without being on the lysosomes.
- 7) VCP aggregation in fly muscle seems to have little relevance other than it is seen in patient muscle. What else co-localizes? ubiquitin, p62, lysosomes, other co-factors?
- 8) the identification of an SVIP variant is intriguing but the genetics for this are not sufficient, no segregation, no other families/patients. It may have helped identify a variant that is dysfunctional in the flies but I am very concerned about using this to bolster human genetics in a single sporadic patient with a heterozygous variant. In addition, CHMP2B, not Chmp4b and was VCP evaluated in this patient.
- 9) are there any thoughts about how the SVIP variant might affect function. It is really just a

missense at the terminal amino acid. Hard to imagine how that would disrupt structure.

10) It is difficult to reconcile how VcP disease could be related to SVIP failed function when only one of several VCP mutations have failed interaction. Something else must be going on. This should be elaborated on in the discussion further.

Reviewer #2 (Remarks to the Author):

Johnson and colleagues have made important, interesting progress on both the SVIP and VCP proteins in cellular and organismal health and disease. This paper is also an important step forward for the autophagy field in understanding how autophagosome fusion with lysosomes, the last step in autophagy, is regulated in vivo.

While prior cell-based work suggested that SVIP can affect autophagy, this study represents several firsts by showing that SVIP affects autophagosome-lysosome fusion in vivo, affects organismal health, and influences muscle and neuron degeneration in vivo. Moreover, the authors show that SVIP is a functional binding protein for VCP, a molecule with extensive links to muscle and neuron degenerative diseases. SVIP is found to mediate VCP recruitment to lysosomes thereby affecting autophagosome-lysosome fusion. Thus, while VCP is known to influence numerous cellular processes, the results here indicate that the VCP/SVIP interaction figures prominently in affecting the late stages of autophagy.

Johnson and colleagues also provide compelling genetic links between SVIP/VCP and degenerative disease. They move from biochemical mechanisms of how disease alleles affect SVIP/VCP binding and lysosomes to functional analysis of a new SVIP mutation found in a patient with FTD. While several binding proteins/cofactors have been identified for VCP, disease links to mutations in both SVIP and VCP studied here suggest that SVIP/VCP regulation of autophagosome-lysosome fusion might be particularly relevant for muscle and neuron degenerative diseases.

It is worth highlighting that the authors maximally leverage the *Drosophila* model system to study molecules, cellular processes and genetic changes involved in degenerative disease. This study is an excellent example of how invertebrate model systems, such as *Drosophila*, are extremely valuable for understanding the molecular and genetic basis of disease.

With relatively modest revisions, I would see this paper as suitable for Nature Communications. Below, I provide specific comments aimed at improving what is already an elegant, comprehensive piece of science.

Major Comments>>

1) The work on heterozygous VCPP134L animals is an important part of this paper. It would be helpful to test anatomically whether neuron degeneration is occurring in the VCPP134L heterozygous animals. While I agree that electrophysiology data are consistent with impaired neuron function, anatomical data would strengthen the case. The reviewer is keeping in mind that much of the data in the paper shows effects in VCP and SVIP mutants come from muscle.

Given the importance of the electrophysiology data in Figure 8C, quantitation (both frequency and amplitude) and statistics for this data would be valuable. This would provide even more support for several of the authors statements, such as: "The successes are full-blown muscle depolarizations of invariant amplitude, indicating that failures do not originate at the NMJ."

2) In Figure 1E, VCP-sfGFP is only present in the muscle nucleus. Previous work from the Davis lab

(Johnson, 2015) showed that VCP colocalizes with autolysosomes throughout the muscle. It would be helpful if the authors could explain the differences here.

How does overexpression of SVIP with mutations that prevent binding to VCP affect VCP localization?

3) Lifespan and geotaxis data need accompanying statistical analysis throughout the paper (Figure 3, S4 and 8D,E).

Minor Comments>>

1) Figure 2F features a clever and interesting experiment. The data does indicate that FKBP dimerization of VCP with Spin increases VCP localization to lysosomes in the absence of SVIP. However, it is also notable that the general levels of VCP look much higher in the presence of dimerizer/rapamycin. Could the authors comment on this? They might consider including a western blot to examine the overall levels of VCP in the absence and presence of dimerizer.

2) "Next, we generated a Drosophila SVIPS82L transgene. When SVIPS82L is over-expressed in WT muscles, tubular lysosomal architecture is disrupted (Figure 9C)."

Is this statement correct? The genotype labels in Figure 9C look like this experiment is actually SVIP heterozygous animals with transgenic overexpression of wt SVIP or SVIPS82L transgenes.

"The defects are even more pronounced when the SVIPS82L mutation is expressed in an SVIP heterozygous knockout (SVIPKO/+). Lysosomal tubules are no longer visible, and dynamics are severely perturbed (Figure 8C; Supplemental Movies 1 and 2)."

Likewise, is there a correction needed here? I did not see this data in Figure 8C, which is the VCP data. Perhaps the authors have forgot to include some data in Figure 9C?

Given the reviewer's confusion here, would the authors also please confirm the genotype is correct for Supplemental Movie 2.

3) Why do the authors quantify synapse length in Figure 4F? Why not simply quantify the number of Brp puncta? Is there a reason for this?

The effect shown in images (Fig 4A versus 4B) looks closer to a 10 fold decrease in Brp puncta numbers, while quantitation in Fig 4F is closer to a 3 fold effect. Could a more representative image could be shown for Fig 4B, or perhaps this is simply a consequence of how quantitation is done for synapse length instead of using Brp puncta numbers.

4) Figure 3A and E show differences in wt lifespan of 70 and 90 days. Is this correct? If so, the effects in Fig 3E are milder than the variability in wt lifespans across different experiments. The reviewer suggests caution here, and consideration of whether data in 3E should be removed.

5) In Figure 2B, the quantitative data on tubules/100um shows that rescue with SVIP RR is not significantly different from wt. Is this correct? The SVIP RR rescue clearly does not look like it is rescuing, but the way the stats are done it argues against this. The authors might consider comparing SVIP RR rescue data with SVIP knockout and wt SVIP rescue data for both datasets.

Given that CRISPR is used to generate the original SVIP allele, it might also be helpful to annotate the rescue data more clearly for wt SVIP rescue and SVIP RR rescue in the genotype labels for this Figure.

6) Figure 4D appears to be missing some significance labels.

7) Are the labels on y-axis in Figure 4D and 4E correct? 4D shows major effects but y-axis is labeled as intensity. 4E shows no effects, but y-axis is labeled as density. Given images shown this might be reversed. To the reviewer, it looks like density/number of Dlg puncta is way down in aged SVIP animals, but the levels of the Dlg staining where puncta are present is normal.

Reviewer #3 (Remarks to the Author):

The manuscript by Johnson et al, entitled 'SVIP is a Molecular Determinant of Lysosomal Dynamic Stability, Neurodegeneration and Lifespan' is a very well written, logical manuscript. The work is a direct continuation of the lab's previous paper showing the role for VCP in maintaining the tubular lysosomal network in muscle. In the manuscript submitted here, they have identified that the VCP-interacting protein SVIP specifically recruits VCP to lysosomes and is required for proper function (as indicated by defective autophagy and mitophagy(?)). The authors convincingly demonstrate the physiological importance of SVIP-VCP to muscle and neuromuscular function, with clear implications for neurodegeneration by using knock-out of SVIP, as well as VCP and SVIP mutations that prevent SVIP-VCP binding and a FTD patient SVIP mutation. Overall the data are well presented, logical and significantly advance our understanding of the mechanisms of disease caused by SVIP-VCP disruption.

With a few amendments, I would recommend the manuscript for publication.

Specific comments:

- Does neuron-specific expression of SVIP rescue defects seen in Figure 6?
- There is little mechanistic insight into the S82L mutation; how does this mutation disrupt lysosomes and is VCP still recruited to the lysosome? If not, do you see the same phenotype when you treat flies with tunicamycin as seen in Fig. 3H?

Minor comments:

- Figure 1C, what is the significance of the 2 mutated arginines?
- Labels in 2B are difficult to see
- Figure 3H; the authors postulate that loss of SVIP may be protective in tunicamycin-dependent ER stress because there is more VCP available. Could the authors look at VCP localization in muscle of tunicamycin-treated flies, is it recruited to ER?

RESPONSE TO REVIEWER CRITICISM

First, we would like to thank all of the reviewers for their generally supportive comments and enthusiasm. We briefly summarize this enthusiasm for all reviewers and editors.

Reviewer 1 states, *“Overall, the manuscript and study is interesting as little is understood about SVIP function and VCP co-factor specific functions are only beginning to be teased out.”*

Reviewer 2 states, *“Johnson and colleagues have made important, interesting progress on both the SVIP and VCP proteins in cellular and organismal health and disease. It is worth highlighting that the authors maximally leverage the Drosophila model system to study molecules, cellular processes and genetic changes involved in degenerative disease. This study is an excellent example of how invertebrate model systems, such as Drosophila, are extremely valuable for understanding the molecular and genetic basis of disease. With relatively modest revisions, I would see this paper as suitable for Nature Communications.”*

Reviewer 3 states, *“Overall the data are well presented, logical and significantly advance our understanding of the mechanisms of disease caused by SVIP-VCP disruption. With a few amendments, I would recommend the manuscript for publication.”*

Second, we would like to thank the reviewers for their insightful criticism. We have spent the past six months addressing each of the comments of each of the three reviewers. In most cases, we have been able to directly address comments and criticisms through the addition of new data and new quantification of existing data.

Our response to reviewer criticism is organized as follows: On the next page of our response, we summarize the full extent of new data that we have added to our manuscript. Following that summary, we specifically respond to each comment of each reviewer.

We are particularly grateful for the time and energy necessary to review our work given all of the extraordinary challenges that laboratories and individuals are facing during the current pandemic. We also would like to similarly thank the editors for their continued efforts and support. We wish everyone the best of health.

Sincerely,

Graeme Davis
(on behalf of all authors of this study)

SUMMARY OF NEWLY ADDED DATA AND ANALYSES.

1. We have added numerous new experiments and analyses to **Figure 1** in response to reviewer questions and criticism. Specifically, we provide new quantitation demonstrating a strong correlation of endogenously tagged SVIP protein with the tubular lysosomal network. Similarly, we provide new data demonstrating the presence of endogenously tagged VCP on tubular lysosomes. Then, we provide additional quantitative evidence that overexpression of SVIP is sufficient to re-localize endogenously tagged VCP protein to the lysosomal network. Finally, we provide new evidence that the *SVIP^{RR}* mutant, which does not bind VCP, is unable to achieve VCP re-localization when overexpressed. All of the effects are quantified and are highly statistically significant.

2. We have fully revised and extended **Figure 3**. New statistical analyses are applied to all data sets, demonstrating highly significant effects in all instances. Geotaxis behavioral assays were repeated and analyzed to allow appropriate statistical comparisons. None of the original conclusions were altered, and all are now supported by statistical arguments.

3. As requested, we have added another quantitative analysis of a muscle marker to **Figure 4** demonstrating disruption of muscle integrity. These data complement our existing light-level and ultrastructural analyses. We have also updated terminology to make our analyses more clear.

4. As requested, we have added quantification to our ultrastructural analyses. These data are now presented in **Figure 5**.

5. In **Figure 6**, we have added a substantial amount of new data to address reviewer questions and concerns. This includes additional rescue experiments and additional electrophysiological analyses.

6. In **Figure 6** we have added an entirely new data set, examining cell death in the nervous system. We demonstrate that aged *SVIPKO* animals have elevated cell death throughout the CNS. Remarkably SVIP expression in muscle is sufficient to rescue cell death in regions of the central nervous system that are dominated by motor circuitry.

7. In **Figure 8**, we have added quantification of the electrophysiological results, as requested. We have also added new data regarding anatomical degeneration.

8. We have revised the text of our manuscript to reflect the addition of these new data and analyses. Text revisions were also made to directly address the concerns of the reviewers.

In summary: We have made every effort to directly address reviewer comments through the addition of new data, new analyses and text revision. We feel that our manuscript has been greatly strengthened as a result of these modifications and would like to thank the reviewers, once again, for their time and input.

RESPONSE TO QUERIES FROM INDIVIDUAL REVIEWERS

Reviewer #1 (Remarks to the Author):

The manuscript by Johnson et al explores the effects of SVIP KO, a VCP cofactor, in *Drosophila*. Specifically, they find that SVIP recruits VCP to tubular lysosomes and loss of SVIP phenocopies VCP KO from their previous paper and to some extent a VCP mutation that effects SVIP binding. They then further suggest that SVIP may be involved lysosomal proteostasis and even participate in a non cell autonomous aspect of protection. Finally, they identify a single patient and variant in a patient with sFTD and find that this variant disrupts lysosomal function. Overall, the manuscript and study is interesting as little is understood about SVIP function and VCP co-factor specific functions are only beginning to be teased out. However, the study is extensive leading to several areas that are unconvincing.

Query #1. In Figure 1, how do we know for sure that SVIP is on lysosomes. I appreciate that spin is a lysosomal protein but it might be nice to have another marker that confirms lysosomal localization. Figure 1E, wouldn't it be more important to quantitate the amount of VCP on lysosomes (SPIN colocalization) rather than loss from the nucleus.

RESPONSE: We appreciate this suggestion. In the original Figure 1D, we presented representative images showing co-localization of endogenously tagged SVIP protein with the distributed lysosomal network. To complement these original experiments, we have added a new data set demonstrating that endogenous SVIP shows near-perfect co-localization with tubular lysosomes, an effect that is established with an unbiased, image-based co-localization analysis with statistical measures (Manders correlation coefficients were calculated for both imaging channels that we assessed). Finally, we now extend our analyses by demonstrating that over-expression of the mutant SVIP (SVIP^{RR}), which does not bind VCP, fails to re-localize endogenously tagged VCP to the lysosome. In these new data, we quantify re-localization to the lysosome as requested. These data complement our prior analyses demonstrating re-localization away from the nucleus.

Finally, we would like to point out that the endogenous tagging of VCP with GFP is a novel approach in the current work. We do not make a major point of this in the paper, but want to bring this to the attention of the reviewers as it impacts direct comparisons with prior localization work (ours and others) that was done with over-expression of VCP.

Query #2) Figure 2 demonstrates that VCP can be recruited in an artificial system. Why is VCP in the endogenous state not on the lysosome in control flies. It seems to only be present when SVIP is overexpressed (Figure 1E) or when VCP:SVIP co-localization occurs. Where is basal VCP what recruits it normally to lysosomes?

RESPONSE: We appreciate this comment and we have adjusted the text to make our experiment more clear. The experiment in question (Figure 2E-F) is performed *in the SVIP^{KO} muscles where VCP is no longer recruited to lysosomes*. The lack of VCP at the lysosome is predicted and is precisely the point of the experiment. The baseline condition in the SVIP^{KO} lacks VCP on lysosomes. Activation of the dimerization system is sufficient to recruit VCP to the tubular lysosomes in the absence of SVIP and this is sufficient to restore the tubular architecture of the lysosomes. The point of the experiment is to demonstrate that VCP recruitment to the lysosomes is sufficient sustain normal lysosomal architecture in the absence of SVIP. We acknowledge that the logic of the experiment may not have been described clearly enough in the text, and have made revisions to state experimental system and expectations more clearly. We also hope that the addition of new data to figure 1 will help further clarify.

Query #3) It is essential to demonstrate that bafilomycin is truly effecting autophagy at the dose used. More importantly and significant problem is the treatment with tunicamycin. What is the level of ER stress that is present already in the SVIP KO. The claim that is made about SVIP loss leading to enhanced proteostasis via ER stress is not well justified and relies on this experiment alone.

RESPONSE: We have made substantial alterations to this figure and the associated text in response to this comment.

First, the concentrations of bafilomycin and tunicamycin that we used were based on previously published work, precisely for the purpose shown. We now cite these papers and we apologize for the omission.

Second, we have performed a western blot analysis to examine GRP78/BiP levels and observe an increase in GRP78 levels in flies fed tunicamycin (both WT and SVIP^{KO}) (see new supplemental figure; Figure S4C). Thus, the dose of tunicamycin we use in the lifespan feeding assay is sufficient to induce ER stress. Moreover, we observe that the lifespan of WT animals is cut in half, indicating that the drug is having a major effect on animal physiology. We do not know the organ(s) specifically affected, but we can say with confidence that the known action of the drug is having an effect at the dose given.

Third, we have added substantial statistical analyses to all of the data presented in Figure 3. All of the observed effects are highly statistically significant.

Finally, the reviewer is correct that we have not directly tested this in our study. However, to be fair, the potential effects of loss of SVIP on ER proteostasis is based on previously published work, which we previously cited. It is unusual for a knockout animal to survive better (dramatically so) in the presence of tunicamycin concentrations that cut normal lifespan by more than 50%. Given our data, previously published information, and the known action of tunicamycin (supported by new Western in supplemental information), we think that our text reflects the appropriate combination of caution and explanation. We state, "*Several possible explanations exist. Among these, SVIP^{KO} animals could be Tunicamycin resistant because increased amounts of VCP can be deployed to the ER in the absence of SVIP-mediated localization of VCP to the lysosome. This is consistent with the initial identification of SVIP as a negative regulator of ER associated protein degradation²⁵.*"

Query #4) Figure 4 muscle morphology is intriguing however, it is unclear why more traditional markers are not used focusing on structural elements. DLG is a t tubule marker and may be altered with lysosomal pathology as it is within the vesicular/endomembrane system. The EM helps but would like to see a different marker of morphology.

RESPONSE: We thank the reviewer for this suggestion. First, it seems important to emphasize the importance and relevance of our existing EM analysis. The reason that we pursued the electron microscopy is that it reveals ALL of the membrane systems within the muscle cell and does so in a manner that is independent of the localization of any individual protein marker. We have more than 20 years experience with muscle EM in *Drosophila* and the images were so extraordinary that we initially felt that the representative images were sufficient. *This was clearly an error that we have corrected.* We performed the EM on multiple animals and have now quantified the effects on synaptic membranes (vesicles/vacuole diameter), muscle dimensions and mitochondrial voids. In

all cases the effects are highly significant, as indicated by the representative images. In most instances, this type of high quality EM is considered a gold standard.

Second, after further consideration, we appreciate that this reviewer's concern might be shared by other readers, and we have added the requested additional quantitative analysis of another muscle marker to further highlight the extent of muscle disruption. These data are now presented in a fully revised and updated Figure 5 and 6. Also note that we have added a new quantification of the Dlg marker using additional image-based, unbiased metrics. The effects on Dlg, as a t-tubule marker, are even more dramatic than initially described.

Query #5) The focus on a non-cell autonomous aspect of SVIP functionality on degeneration and subsequent rescue is interesting but again, no evaluation of neuronal pathology and relies on a single set of experiments reduces enthusiasm and is again where this manuscript takes a detour from a straightforward story.

RESPONSE: In general, we consider this a reasonable criticism of our work (see below for addition of new data). However, we think it important to rebut the notion that these analyses are a 'detour'. We previously provided an in-depth analysis of neuronal degeneration in the SVIP knockout, inclusive of timed analysis of dendrite development and morphology in supplemental data, as well as a detailed morphological analysis of motoneuron morphology that was supported by electrophysiological deficits. Thus, we previously provided numerous lines of evidence to document the effects of SVIP knockout on central neuronal maintenance and function. This is central to the phenotypic characterization of the SVIP knockout. We argue that our previously presented data were extensive and by no means a 'detour'.

However, we do acknowledge that our data examining **muscle rescue** of the SVIP knockout included only an analysis of motoneuron function and animal viability (see supplemental Figure S4 as originally presented). We felt we could do more. Therefore, we have pursued an extensive analysis of cell death in the central nervous system of the adult. We demonstrate that there is a highly significant increase in neuronal cell death in the adult CNS in the aged SVIP knockout compared to controls. We also demonstrate that re-supply of SVIP in muscle is able to fully rescue the cell death phenotype in the central nervous system. Notably, the rescue of neuronal cell death is highly specific to the region of the central nervous system that include the pools of neurons (motoneurons and associated interneurons) that innervate the major muscle groups of the abdomen, appendages and flight muscles. Muscle specific expression of SVIP does not rescue neuronal cell death in the *Drosophila* head or visual system. Thus, the effects of muscle-specific rescue appear selective to those neuronal populations that are closely associated with the major muscle groups of the organism.

We thank the reviewer for prompting us to perform this experiment. We feel that this experiment, in particular, has greatly strengthened the impact and importance of the data set as a whole.

Query #6) Figure 7. I would like to know what are the comparable human mutations. Is the A154R supposed to correspond the G157R mutation. What is the A291E in humans. Can the biochemistry be quantitated? It is difficult to judge the magnitude of the non P134L mutations. For 7H, where is VCP? is on Spin RFP structures. Or is it just affecting tubulation without being on the lysosomes.

RESPONSE: There are several questions here – each are addressed.

- A) We apologize for confusion regarding the human mutations. We also apologize that there was a typo in the previous draft regarding the A291E mutation, which should be A229E (A232E in humans). This has been corrected in text and figure. We have also included a new supplemental table listing the corresponding human mutations.
- B) The biochemistry was quantified as originally presented. Binding affinities are determined and additional quantification presented in F and G.
- C) For figure 7H, the representative images show the phenotype, which is the intended purpose of the experiment. We acknowledge that more information regarding VCP is generally important and the information that we now provide in figure 1 will hopefully be sufficient to underscore the presence of VCP at the tubular lysosomes. Prior data in this paper and previously published data from our earlier paper (Johnson et al., 2015) demonstrates that loss of VCP from the tubular lysosome (even when achieved acutely during live imaging) causes disruption of the tubular lysosomal system. So, the phenotype that we show is diagnostic of the loss of VCP from this site. Likewise, the preponderance of information that we have presented, including use of our synthetic dimerization system and SVIP knockout characterization supports this conclusion.

None-the-less, we have added new experimental data to the study in response to this query (Figure S7). We present co-imaging of VCP^{P134L}-GFP and Spin-RFP and present image-based, unbiased quantification demonstrating that VCP^{P134L}-GFP does not co-localize with the tubular lysosome, instead localizing diffusely throughout the cytoplasm. These additional imaging experiments further support our conclusions that the P134L mutation blocks VCP-SVIP binding and prevents VCP recruitment to lysosomes, resulting in the observed lysosomal phenotype. We thank the reviewer for prompting us to do this additional work for our study.

Query #7) VCP aggregation in fly muscle seems to have little relevance other than it is seen in patient muscle. What else co-localizes? ubiquitin, p62, lysosomes, other co-factors?

RESPONSE: We refer the reviewer to our prior publication that focused exclusively on the function of VCP protein in muscle, with numerous co-localization experiments including dynamic live imaging experiments that directly address this issue. Further, in the prior paper we demonstrate co-localization of the aggregates with Ubiquitin (Johnson et al., 2015).

Query #8) the identification of an SVIP variant is intriguing but the genetics for this are not sufficient, no segregation, no other families/patients. It may have helped identify a variant that is dysfunctional in the flies but I am very concerned about using this to bolster human genetics in a single sporadic patient with a heterozygous variant. In addition, CHMP2B, not Chmp4b and was VCP evaluated in this patient.

RESPONSE: We agree that abundant caution is warranted. But, we do not think it either fair or correct to suggest that the only value of the human mutation is to identify a fly mutant to study. With all due respect, that is overly critical.

The value that we see in the human data is that it highlights *potential* relevance of SVIP function the human condition, and it highlights the importance of autolysosomal fusion mediated by SVIP as being potentially relevant to neurodegenerative disease. The SVIP gene is very small and, as such, is unlikely to represent a substantial disease burden. But, by reporting the information, we hope that some readers will be encouraged to look within their data for SVIP or SVIP related information that might be relevant. We have revisited the text of our manuscript to ensure appropriate caution. We would like to point out that we

stated in our original introduction section, that we do not claim causality. We wrote, “...and we present the identification of a novel SVIP mutation in a human patient diagnosed with fronto-temporal dementia (FTD). We subsequently confirm the pathogenicity of this new human SVIP mutation in our *Drosophila* model. Although we cannot conclude causality, this human mutation and our subsequent characterization in vivo in *Drosophila* argues that the function of SVIP has relevance to the human condition.”

The paragraph written in our discussion section also highlights our intent to encourage future investigation, but with appropriate caution regarding causality. “Finally, we identify a human patient with FTD harboring a mutation in SVIP and demonstrate that this mutation causes pathological defects in *Drosophila* muscle lysosomes and impairs autophagy that parallel the phenotypes of our other *Drosophila* disease models. The fact that SVIP is a small gene makes it unlikely that a substantial patient population will harbor SVIP mutations, possibly precluding determination of whether SVIP mutations are actually causal for human disease. None-the-less, the fact that the patient derived mutation is pathogenic in *Drosophila* muscle, phenocopying other disease models, argues that SVIP may be relevant to the human condition and encourages further investigation into the potential benefits of SVIP over-expression as potential suppressor of human disease.”

Query #9) are there any thoughts about how the SVIP variant might affect function. It is really just a missense at the terminal amino acid. Hard to imagine how that would disrupt structure.

RESPONSE: We agree that the we do not know how mutation of the C-terminal serine will affect function. Yet, our experimental evidence clearly demonstrates that it does. Additionally, in response to another reviewer’s suggestion, we have added lifespan data on tunicamycin to our paper demonstrating that the *SVIP*^{S82L} mutant not only disrupts the tubular lysosomal network, but also phenocopies the *SVIP*^{KO} in this regard.

Query #10) It is difficult to reconcile how VCP disease could be related to SVIP failed function when only one of several VCP mutations have failed interaction. Something else must be going on. This should be elaborated on in the discussion further.

RESPONSE: The myriad functions of VCP in the cell were highlighted in the introduction to the paper and this was the main motivation for initiating this work. Our study demonstrates that SVIP participates in one aspect of VCP function. Our original model of VCP dysregulation is consistent with the known literature. Disrupting the cellular balance of VCP activity could have far reaching disease-related consequences. Thus, we generally agree with the assertion that it is unlikely that every disease mutation will cause disease in precisely the same way.

Reviewer #2 (Remarks to the Author):

Johnson and colleagues have made important, interesting progress on both the SVIP and VCP proteins in cellular and organismal health and disease. This paper is also an important step forward for the autophagy field in understanding how autophagosome fusion with lysosomes, the last step in autophagy, is regulated in vivo.

While prior cell-based work suggested that SVIP can affect autophagy, this study represents several firsts by showing that SVIP affects autophagosome-lysosome fusion in vivo, affects organismal health, and influences muscle and neuron degeneration in vivo. Moreover, the authors show that SVIP is a functional binding protein for VCP, a molecule with extensive links to muscle and neuron degenerative diseases. SVIP is found to mediate VCP recruitment to lysosomes thereby affecting autophagosome-lysosome fusion. Thus, while VCP is known to influence numerous cellular processes, the results here indicate that the VCP/SVIP interaction figures prominently in affecting the late stages of autophagy.

Johnson and colleagues also provide compelling genetic links between SVIP/VCP and degenerative disease. They move from biochemical mechanisms of how disease alleles affect SVIP/VCP binding and lysosomes to functional analysis of a new SVIP mutation found in a patient with FTD. While several binding proteins/cofactors have been identified for VCP, disease links to mutations in both SVIP and VCP studied here suggest that SVIP/VCP regulation of autophagosome-lysosome fusion might be particularly relevant for muscle and neuron degenerative diseases.

It is worth highlighting that the authors maximally leverage the *Drosophila* model system to study molecules, cellular processes and genetic changes involved in degenerative disease. This study is an excellent example of how invertebrate model systems, such as *Drosophila*, are extremely valuable for understanding the molecular and genetic basis of disease.

With relatively modest revisions, I would see this paper as suitable for Nature Communications. Below, I provide specific comments aimed at improving what is already an elegant, comprehensive piece of science.

Major Comments

Query #1A) The work on heterozygous VCPP134L animals is an important part of this paper. It would be helpful to test anatomically whether neuron degeneration is occurring in the VCPP134L heterozygous animals. While I agree that electrophysiology data are consistent with impaired neuron function, anatomical data would strengthen the case. The reviewer is keeping in mind that much of the data in the paper shows effects in VCP and SVIP mutants come from muscle.

RESPONSE: While we agree that the *VCP*^{P134L} mutant is important, we have focused our efforts on demonstrating that muscle specific rescue of the SVIP knockout restores both muscle integrity and neuronal integrity in the central nervous system (see new data added to Figures 6 and 8). None-the-less, we have added two additional data points, as requested. First, we demonstrate that the NMJ deteriorates based on loss of ant-BRP labeled active zones in the *VCP*^{P134L} mutant (new data Figure 8G, H). This is consistent with the SVIP knockout phenotype. Second, we have quantified and extended our electrophysiological analyses demonstrating that the motor axons are present but there is failure to recruit the motor neurons, consistent with the observed anatomical dendritic defects observed in the SVIP knockout (Figure 8D).

Query 1B) Given the importance of the electrophysiology data in Figure 8C, quantitation (both frequency and amplitude) and statistics for this data would be valuable. This would provide even more support for several of the authors statements, such as: “The successes are full-blown muscle depolarizations of invariant amplitude, indicating that failures do not originate at the NMJ.”

RESPONSE: We thank the reviewer for requesting these data. We have added this quantification, as mentioned above. We have also assessed the integrity of the motor axons both functionally and anatomically. The data are consistent with our initiation conclusion that there is failure to recruit the motor axons, and when the motor axons are recruited, they are able to convey information to the muscle cell via NMJ that are present, though showing signs of degeneration (decreased Brp puncta number).

Query #2) In Figure 1E, VCP-sfGFP is only present in the muscle nucleus. Previous work from the Davis lab (Johnson, 2015) showed that VCP colocalizes with autolysosomes throughout the muscle. It would be helpful if the authors could explain the differences here.

RESPONSE: We appreciate this suggestion, and it is one that was also posed by Reviewer #1. We copy our response to reviewer #1 here.

In the original Figure 1D, we presented representative images showing co-localization of endogenously tagged SVIP protein with the distributed lysosomal network. To complement these original experiments, we have added a new data set demonstrating that endogenous SVIP shows near-perfect co-localization with tubular lysosomes, an effect that is established with an unbiased, image-based co-localization analysis with statistical measures (Manders correlation coefficients were calculated for both imaging channels that we assessed). Finally, we now extend our analyses by demonstrating that over-expression of the mutant SVIP (*SVIP^{RR}*), which does not bind VCP, fails to re-localize endogenously tagged VCP to the lysosome. In these new data, we quantify re-localization to the lysosome as requested. These data complement our prior analyses demonstrating re-localization away from the nucleus.

Query #3) How does overexpression of SVIP with mutations that prevent binding to VCP affect VCP localization?

RESPONSE: We would like to thank this reviewer for prompting us to pursue an additional experiment. When we over-express the *SVIP^{RR}* mutation that does not bind VCP, we observe that VCP does not re-localize to the lysosomes. These new data are fully quantified and presented in a revised Figure 1.

Query #3) Lifespan and geotaxis data need accompanying statistical analysis throughout the paper (Figure 3, S4 and 8D,E).

RESPONSE: We are grateful that this reviewer prompted us to explore the statistical analysis of lifespan data. We now present full statistical analyses of all lifespan data throughout the paper. All statistical values are now presented with each graph in figure format.

Minor Comments>>

MINOR Query #1) Figure 2F features a clever and interesting experiment. The data does indicate that FKBP dimerization of VCP with Spin increases VCP localization to lysosomes in the absence of SVIP. However, it is also notable that the general levels of VCP look much higher in the presence of dimerizer/rapamycin. Could the authors comment on this? They might consider including a western blot to examine the overall levels of VCP in the absence and presence of dimerizer.

RESPONSE: A related question was posed by reviewer number 1, prompting us to adjust this section of the text to make the experiment more clear. The important point is that the experiment is conducted in the *SVIP* knockout background. In the absence of SVIP chaperone, VCP is not recruited to the lysosomes. Visually, the absence of a structured localization of VCP makes it appear absent. Once the dimerizer is initiated, VCP is recruited to the lysosomes, concentrating on a cellular structure that is easily visualized. The representative images are indicative of the re-localization event that occurs in the complete absence of the SVIP protein. We hope that our revised text makes this more clear.

MINOR Query #2) “Next, we generated a *Drosophila* SVIPS82L transgene. When SVIPS82L is over-expressed in WT muscles, tubular lysosomal architecture is disrupted (Figure 9C).

Is this statement correct? The genotype labels in Figure 9C look like this experiment is actually SVIP heterozygous animals with transgenic overexpression of wt SVIP or SVIPS82L transgenes. “The defects are even more pronounced when the SVIPS82L mutation is expressed in an SVIP heterozygous knockout (SVIP^{KO}/+). Lysosomal tubules are no longer visible, and dynamics are severely perturbed (Figure 8C; Supplemental Movies 1 and 2).” Likewise, is there a correction needed here? I did not see this data in Figure 8C, which is the VCP data. Perhaps the authors have forgot to include some data in Figure 9C?

RESPONSE: The reviewer is correct, the representative images show the effect of SVIP^{S82L} when expressed on the heterozygous SVIP mutant background. We have changed the quoted sentence to read, “*When SVIP^{S82L} is over-expressed in an SVIP heterozygous knockout (SVIP^{KO}/+) background we find that lysosomal tubules are no longer visible and lysosomal dynamics are severely perturbed (Figure 9C; Supplemental Movies 1 and 2).*” We thank the reviewer for pointing this out.

We would like to emphasize that, in our experiments, one normal copy of the *SVIP* gene is still present, reflecting the fact that one wild type copy of the *SVIP* gene would also be present in the setting of autosomal dominant disease. In many experimental systems, the approach would be to rescue the null mutation with the mutant. However, we are able to show a full phenotype when one copy of the normal SVIP protein is present.

We would also like to point out that we have added more data to the paper regarding the SVIP^{S82L} knockin mutant. We now demonstrate that SVIP^{S82L} mutants phenocopy SVIP^{KO} lifespan under conditions of cellular stress, compared to similarly stressed controls.

MINOR Query #3) Why do the authors quantify synapse length in Figure 4F? Why not simply quantify the number of Brp puncta? Is there a reason for this?

RESPONSE: This was simply a choice of metric. In some instances, the Brp puncta at these synapses are difficult to resolved into discrete spots. The images show some of the best images of clearly resolved Brp. Ultimately, the EM provides the nicest information that

is consistent with the actual degeneration of the synaptic terminals including the presence of large vacuoles.

MINOR Query #4) Figure 3A and E show differences in wt lifespan of 70 and 90 days. Is this correct? If so, the effects in Fig 3E are milder than the variability in wt lifespans across different experiments. The reviewer suggests caution here, and consideration of whether data in 3E should be removed.

RESPONSE: Although the reviewer is correct regarding comparison among the graphs, it is not appropriate to make comparisons across the graphs due to the different genotypic backgrounds in each experiment. In each analysis, the controls are genotypically matched to the mutants that we are quantifying. Specifically, in 3A, the 'WT' strain used is w^{1118} and in 3E, the control used is *mhc-GAL4/+* (expressing GAL4 constitutively which may be a source of altered lifespan). Thus, within each graph, the statistically significant differences are correctly controlled, even if the absolute lifespan of each genotypic background may be moderately different.

MINOR Query #5A) In Figure 2B, the quantitative data on tubules/100um shows that rescue with SVIP RR is not significantly different from wt. Is this correct? The SVIP RR rescue clearly does not look like it is rescuing, but the way the stats are done it argues against this. The authors might consider comparing SVIP RR rescue data with SVIP knockout and wt SVIP rescue data for both datasets.

RESPONSE: We thank the reviewer for raising this issue and we have revised how the statistics are shown for this figure. In fact, the line in question was mis-drawn. The new presentation of the stats on the figure should make the data more easily interpretable. For the reviewer's information, the RR mutant is significantly different from wild type ($p < 0.01$), confirming the reviewer's impression, as the data appear to the eye. We apologize for the error and thank the reviewer for the keen eye to pick this up and correct.

MINOR Query #5B) Given that CRISPR is used to generate the original SVIP allele, it might also be helpful to annotate the rescue data more clearly for wt SVIP rescue and SVIP RR rescue in the genotype labels for this Figure.

RESPONSE: Good suggestion. We have updated the labels accordingly.

MINOR Query #6) Figure 4D appears to be missing some significance labels.

RESPONSE: Yes! Fixed.

MINOR Query #7) Are the labels on y-axis in Figure 4D and 4E correct? 4D shows major effects but y-axis is labeled as intensity. 4E shows no effects, but y-axis is labeled as density. Given images shown this might be reversed. To the reviewer, it looks like density/number of Dlg puncta is way down in aged SVIP animals, but the levels of the Dlg staining where puncta are present is normal.

RESPONSE: The labels for the figures were not as informative as they should have been and we apologize for the resulting confusion. Figure 4D previously reported the integrated Dlg intensity across the muscle field. This is quantitative information that matches the representative images above. The previously reported Figure 4E documented the pixel intensity for Dlg staining that remained in the image field (the scattered, infrequent puncta). In retrospect, this information is not particularly useful and we have removed those data from the figure. Instead, we now present an entirely new analysis, at the request of reviewer #1. Here we analyze another reporter, phalloidin, to show the

disruption of the muscle organization. We also show another quantification for the Dlg staining that demonstrates more clearly that the Dlg is not only lost from the muscle field, but generally loses the punctate organization that is expected from localization to the t-tubule system. We think that the addition of these new data make the phenotypic characterization more clear, and more thorough.

Reviewer #3 (Remarks to the Author):

The manuscript by Johnson et al, entitled 'SVIP is a Molecular Determinant of Lysosomal Dynamic Stability, Neurodegeneration and Lifespan' is a very well written, logical manuscript. The work is a direct continuation of the lab's previous paper showing the role for VCP in maintaining the tubular lysosomal network in muscle. In the manuscript submitted here, they have identified that the VCP-interacting protein SVIP specifically recruits VCP to lysosomes and is required for proper function (as indicated by defective autophagy and mitophagy(?)). The authors convincingly demonstrate the physiological importance of SVIP-VCP to muscle and neuromuscular function, with clear implications for neurodegeneration by using knock-out of SVIP, as well as VCP and SVIP mutations that prevent SVIP-VCP binding and a FTD patient SVIP mutation. Overall the data are well presented, logical and significantly advance our understanding of the mechanisms of disease caused by SVIP-VCP disruption.

With a few amendments, I would recommend the manuscript for publication.

Major Comments

Major Query #1) Does neuron-specific expression of SVIP rescue defects seen in Figure 6?

RESPONSE: This is an excellent question. We have added new data to address this issue to Figure 6. Neuronal expression of SVIP does not rescue, only muscle-specific expression achieves rescue.

Major Query #2) There is little mechanistic insight into the S82L mutation; how does this mutation disrupt lysosomes and is VCP still recruited to the lysosome? If not, do you see the same phenotype when you treat flies with tunicamycin as seen in Fig. 3H?

RESPONSE: Although we do not have significant insight into the molecular mechanism of the S82L mutation, we have performed the experiment suggested by the reviewer and found that SVIP^{S82L} animals survive better on TM, albeit to a lesser extent than SVIP^{KO} mutants. These data lend further support that this mutation has a physiological impact on animals that is similar to that of the SVIP knockout.

Minor comments:

Minor Query #1) Figure 1C, what is the significance of the 2 mutated arginines?

RESPONSE: VCP binding is prevented. This mutant is now utilized in Figure 1 to demonstrate that overexpression of SVIP^{RR} fails to re-localize VCP to the tubular lysosomal membranes.

Minor Query #2) Labels in 2B are difficult to see

RESPONSE: We have modified the font labels to make them easier to see.

Minor Query #3) Figure 3H; the authors postulate that loss of SVIP may be protective in tunicamycin-dependent ER stress because there is more VCP available. Could the authors look at VCP localization in muscle of tunicamycin-treated flies, is it recruited to ER?

RESPONSE: Given that this is a feeding assay, affecting all tissues, we felt that this experiment would not be sufficiently conclusive and opted not to do this experiment.

REVIEWERS' COMMENTS

Reviewer #1 (Remarks to the Author):

Many of the points have been clarified and extra experiments and details provided. Specifically the effect of non-cell autonomous expression and the aspects of ER stress have been addressed.

I still have significant concerns about presenting human genetic data. I apologize if this seemed to be overcritical. While the connection to a human patient is exciting, this is a single heterozygous variant in a gene not previously reported to be associated with disease in a single patient. Human genetic studies are challenging and deserve scrutiny before publication. There are specific guidelines for reporting new gene associations. In addition, there are other ways to report unclear genetic associations in public databases and through genetic "matchmaking." So I am not suggesting that this data is not worthy of being shared with other researchers, it is just this is not proper format. The implications for patient care are too great.

Reviewer #2 (Remarks to the Author):

The authors have thoroughly and carefully addressed my concerns. They have also added valuable new experiments and data analysis that substantially improve the manuscript. The authors now have my full support and my congratulations on an exceptionally interesting and impactful story.

Minor comment

Are panels for Figure 8 correctly annotated in this statement?

"Finally, we examined the organismal defects associated with the VCPP134L mutation (Figure 8C-E). Lifespan of VCPP134L mutants were modestly, but significantly, reduced compared to VCPWT (Figure 8D, $p < 0.001$) and displayed progressive mobility defects (Figure 8E)."

Figure C and D refer to electrophysiological data. Figure 8F is geotaxis data that is not cited but is, I believe, what is being referenced by the authors.

Reviewer #3 (Remarks to the Author):

The authors have addressed all of my concerns and I am happy to recommend the manuscript for publication.

Reviewer #1 (Remarks to the Author):

Many of the points have been clarified and extra experiments and details provided. Specifically the effect of non-cell autonomous expression and the aspects of ER stress have been addressed.

I still have significant concerns about presenting human genetic data. I apologize if this seemed to be overcritical. While the connection to a human patient is exciting, this is a single heterozygous variant in a gene not previously reported to be associated with disease in a single patient. Human genetic studies are challenging and deserve scrutiny before publication. There are specific guidelines for reporting new gene associations. In addition, there are other ways to report unclear genetic associations in public databases and through genetic "matchmaking." So I am not suggesting that this data is not worthy of being shared with other researchers, it is just this is not proper format. The implications for patient care are too great.

RESPONSE: As requested by the reviewer and editors (in the email that we received) we have removed the human data presented in Figure 9A, B. We removed reference to the patient, and the patient history and diagnosis from the text of our manuscript. However, as suggested by the editors, we have retained the information on how we discovered the variant that we tested (a screen of the data in the MAC at UCSF) and we retain the analysis/characterization of this mutation in *Drosophila*.

Reviewer #2 (Remarks to the Author):

The authors have thoroughly and carefully addressed my concerns. They have also added valuable new experiments and data analysis that substantially improve the manuscript. The authors now have my full support and my congratulations on an exceptionally interesting and impactful story.

Minor comment

Are panels for Figure 8 correctly annotated in this statement?

“Finally, we examined the organismal defects associated with the VCPP134L mutation (Figure 8C-E). Lifespan of VCPP134L mutants were modestly, but significantly, reduced compared to VCPWT (Figure 8D, $p < 0.001$) and displayed progressive mobility defects (Figure 8E).” Figure C and D refer to electrophysiological data. Figure 8F is geotaxis data that is not cited but is, I believe, what is being referenced by the authors.

RESPONSE: We thank the reviewer for this query. This error has been corrected.

Reviewer #3 (Remarks to the Author):

The authors have addressed all of my concerns and I am happy to recommend the manuscript for publication.

We thank the reviewer for the enthusiastic reviews and critical commentary.